# PREDICTION CONSISTENCY TRAINING ENHANCES SUPERVISED LEARNING FOR LEARNING TASKS WITH COMPLEX LABELS

## ABSTRACT

Directly predicting labels from data inputs has been a long-standing supervised learning paradigm. Its trade-off between compression and prediction is studied under the information theory framework e.g. Information Bottleneck, especially in the context of deep learning. It typically assumes that the information content of labels is significantly less than that of data inputs, leading to model designs that prioritize compressing and extracting features from data inputs. In fact, recent supervised learning increasingly faces predicting complex labels, exacerbating the challenge of learning mappings from compressed latent features to high-fidelity label representations. Predictive bottlenecks emerge not only from compression limitations but also from the inherent complexity of feature-to-label transformations. This paper proposes incorporating scheduled label information into the model during training to better learn the prediction consistency mapping, which stems from the consistency mapping concept from generative consistency models. Unlike traditional approaches predicting labels directly from inputs, in this paper, the training of our designed conditional consistency involves predicting labels using inputs and noise-perturbed label hints, pursuing the predictive consistency across different noise steps. It simultaneously learns the relationship between latent features and a spectrum of label information from zero to complete, which enables progressive learning for complex predictions and allows multi-step inference analogous to gradual denoising, thereby enhancing the prediction quality. Experiments on vision, text, and graph tasks show the superiority of our consistency supervised training paradigm, over conventional supervised training in complex label prediction problems. Source code will be made publicly available upon acceptance.

## 1 INTRODUCTION

Supervised learning has long been a cornerstone of machine learning, where models are trained to map input data to corresponding output labels by minimizing prediction error, which measures the discrepancy between the predicted labels and the ground truth labels. This direct label prediction paradigm has been widely applied across various domains, from image classification (Krizhevsky et al., 2012; He et al., 2016; Simonyan & Zisserman, 2014), natural language processing (Vaswani, 2017; Devlin, 2018; Radford, 2018), to structured graph learning (Kipf & Welling, 2016; Veličković et al., 2017; Wu et al., 2022), due to its simplicity and effectiveness in handling large, annotated datasets. In such systems, it is very typical to employ a neural network to directly map the data inputs to labels, with a particular focus on the expressive capacity of (deep) models to compress high-dimensional inputs into latent representations while preserving relevant information for accurate predictions, viewed from an information theory perspective (Tishby et al., 2000; Tishby & Zaslavsky, 2015). This compression is believed to contribute to the generalization ability of deep learning models, particularly in high-dimensional and noisy input scenarios.

This paradigm typically assumes that the labels contain a significantly lower dimensionality and less information than the data inputs, thus guiding model designs toward compressing and extracting relevant features from the input space for efficient prediction (Tishby & Zaslavsky, 2015). The assumption further implies that transforming meaningful latent features to label outputs is relatively straightforward compared to the challenge of extracting expressive features. However, recent advances

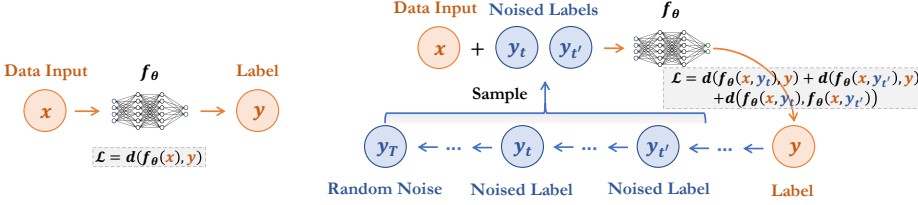

(a) Classic Supervised Learning    (b) Supervised Consistency Learning

Figure 1: Illustration of supervised consistency Learning (SCL). Unlike traditional approaches predicting labels directly from inputs, SCL predicts labels using inputs and noise-perturbed label hints and pursues predictive consistency across different noise steps.

in supervised learning have shown that many modern tasks involve much more complex labels, leading to new challenges. Examples include image prediction extending to dense, pixel-level outputs (Long et al., 2015; Chen et al., 2017), natural language processing tasks generating complex sentences (Brown, 2020; Touvron et al., 2023), and predicting complex structured solutions based on graph representations (Li et al., 2023; Satorras et al., 2021). These challenges reveal predictive bottlenecks beyond feature extraction. To address this challenge, one approach involves learning an efficient representation of the complex labels, facilitating a more effective transformation within the low-dimensional feature space. Indeed, this can correspond to methods that leverage Variational Auto-Encoder (VAE) (Kingma, 2013) to perform learning tasks within the latent space (Rombach et al., 2022; Hottung et al., 2021). However, this approach necessitates that the transformation between labels and latent features be reversible, requiring the training of two additional neural networks (an encoder and a decoder) to accurately capture and reconstruct the label information.

In this paper, we propose an alternative approach, aiming to better capture complex label information by introducing a fundamentally different supervised learning paradigm. We leverage the concept of consistency mapping from generative consistency models Song et al. (2023) to frame the supervised learning process as learning the prediction consistency, transitioning from noised labels to full labels conditioned on the data input. Specifically, we introduce Supervised Consistency Learning (SCL), which establishes trajectories from different noise levels on the target labels to the raw labels conditioned on the given data inputs. This process can be interpreted as a conditional generation mechanism, where high-fidelity labels are inferred from noisy counterparts using the input data as a conditional guide, as shown in Fig. 1. Yet, within the supervised setup, each training instance has a reference target label, and the model learns to guide all denoising trajectories to this label by enforcing this predictive consistency across different noise timesteps, which we define as prediction consistency.

During training, unlike conventional supervised learning predicting labels directly from inputs, SCL maps noisy labels at varying noise levels back to the true label conditioned on the input data and enforces different noise timesteps mapping to the same target. By enforcing predictive consistency across multiple noise levels, the model captures a rich spectrum of label information from entirely noisy to wholly accurate predictions, fostering a more expressive mapping between latent features and labels. During inference, the inherent multi-step denoising mechanism also facilitates progressive refinement, resulting in more flexible and accurate predictions for complex labels. Intuitively, this process can be seen as learning to predict with varying degrees of solution hints, which benefits learning by progressively understanding the label information, especially when the labels are complex.

We demonstrate the effectiveness of our approach across a range of tasks involving complex labels from diverse domains, including vision learning (e.g., semantic segmentation (Long et al., 2015; Chen et al., 2017)), graph learning (e.g., N-body simulation (Satorras et al., 2021) and combinatorial optimization (Li et al., 2023)), and natural language processing (e.g., next-token prediction in large language models (Brown, 2020; Touvron et al., 2023)). The empirical results highlight the superiority of SCL over traditional supervised learning across various mainstream network backbones.

## 2 RELATED WORK AND PRELIMINARIES

**Supervised Learning and Information Interpretation.** In supervised learning (SL), the models typically learn the direct mapping from the data inputs to labels by minimizing a loss function that captures the discrepancy between predicted and ground truth labels. In particular, the Information Bottleneck (IB) principle (Tishby et al., 2000; Tishby & Zaslavsky, 2015) offers a theoretical

framework to analyze (deep) learning systems, indicating that models balance the trade-off between compression and prediction accuracy. The IB method (Tishby et al., 2000) aims to find a compressed representation of the input that retains relevant information about the target while discarding irrelevant details. The IB method formulates this trade-off by minimizing the mutual information between the input and a compressed representation, while maximizing the mutual information between the compression and the target. By considering the relationship between the input and the label through the lens of information theory, IB provides a powerful tool for understanding model generalization and optimizing feature representations in supervised learning.

**Diffusion Models and Consistency Models.** Diffusion models are characterized by a forward process of noise injection and a reverse process of learnable denoising, where neural networks iteratively predict data distributions conditioned on increasingly noisy inputs. In continuous space, these models are closely linked to Stochastic Differential Equations (SDEs), with techniques such as the Probability Flow ODE offering a deterministic approximation to sample generation (Sohl-Dickstein et al., 2015; Song & Ermon, 2019; Ho et al., 2020; Song et al., 2020a; Song & Ermon, 2020; Nichol & Dhariwal, 2021; Dhariwal & Nichol, 2021). Extensions to discrete data have also been explored, with noise distributions modeled as binomial or categorical variables (Sohl-Dickstein et al., 2015; Austin et al., 2021; Hoogeboom et al., 2021). Building on the advancements of diffusion models, consistency models (Song et al., 2023; Song & Dhariwal, 2023) have introduced an alternative paradigm to accelerate the generation process. Rather than iteratively refining noisy samples through a reverse diffusion process, consistency models leverage a self-consistency mechanism across different time steps, directly learning the mappings from noise to data in a single step or a small number of steps. This approach has shown promise in reducing computational overhead while maintaining high sample quality.

## 3 SUPERVISED LEARNING TASKS WITH COMPLEX LABELS

Supervised learning aims to train the model to extract compressed features or representations of input data $\mathbf{x} \in X$ while retaining the most relevant information about the target label $\mathbf{y} \in Y$ (Tishby et al., 2000; Tishby & Zaslavsky, 2015). Note this is based on the assumption that the data provides sufficient information about the labels, which means the data is abundant. From an information-theoretic perspective, the mutual information $I(X, Y)$ quantifies how much information $X$ provides about $Y$. Typically, $X$ is a high dimensional variable of a low-level representation of the data, such as pixels of an image, whereas $Y$ has a significantly lower dimensionality of the predicted categories, which generally means that most of the entropy of $X$ is not very informative about $Y$ and that the relevant features in $X$ are highly distributed and difficult to extract (Tishby & Zaslavsky, 2015). In deep learning (LeCun et al., 2015), deep neural networks create a compressed representation $X_E$ of $X$ through an encoder, which discards irrelevant information while preserving as much of the mutual information $I(X_E, Y)$ as possible. The compression is optimized by minimizing $I(X, X_E)$, the information between $X$ and its compressed form $X_E$, while maximizing $I(X_E, Y)$, the information between the compressed representation and the target. This trade-off can be formalized in the IB objective: $\min_{p(X_E|X)} [I(X, X_E) - \beta I(X_E, Y)]$, where $\beta$ is a Lagrange multiplier that governs the balance. In real-world scenarios, compression is often lossy, meaning that some information about the input signal $X$ is inevitably discarded. Consequently, the challenge becomes ensuring that the model retains only the information about $Y$ that is necessary for the task while minimizing redundancy.

This formulation typically assumes that $Y$ is a low-dimensional vector (e.g., class labels) where the information content is relatively limited. However, many real-world tasks, especially in structured prediction (e.g., image segmentation, sequence generation), involve predicting high-dimensional outputs. In these tasks, the mutual information $I(X_E, Y)$ can be difficult to maximize because the high-dimensional labels themselves contain redundancies, and fitting a model to predict them from $X_E$ becomes a non-trivial task. Moreover, the space of possible outputs $Y$ could involve complex correlations that are hard to capture directly. Below, we formalize such tasks with a qualitative definition.

**Definition 3.1.** A learning task with complex labels is characterized by a label space that exhibits high complexity due to one or more of the following characteristics: (i) high dimensionality, (ii) intricate internal structure, or (iii) the presence of significant dependency patterns among label components.

In contrast to traditional tasks with simple scalar or categorical labels, complex labels encode rich, multi-dimensional, or structured information. Consequently, these tasks require models to capture sophisticated relationships and dependencies within the label space, transcending straightforward mappings from input features. The inherent complexity of the label space suggests the need for

learning an effective latent representation $Y_E$ of the target $Y$. This concept aligns with existing approaches (Rombach et al., 2022; Hottung et al., 2021)that handle high-dimensional outputs in latent spaces leveraging methods like Variational Autoencoders (VAE) (Kingma & Welling, 2014). However, for prediction purposes, these methods rely on the invertibility of the mapping from $Y$ to its latent representation $Y_E$, requiring both an encoder to compress $Y$ and a decoder to reconstruct $Y_E$ back to $Y$ for prediction. This necessitates learning additional networks to manage latent representations. In the following section, we present an alternative approach that enhances the model's ability to capture $I(X_E, Y)$ directly by leveraging the mechanism of the learning paradigm itself, thereby avoiding the need for introducing additional networks and its associated computational overhead.

# 4 THE SUPERVISED CONSISTENCY LEARNING FRAMEWORK

This section presents the proposed supervised consistency learning framework. We begin by introducing the diffusion trajectories for labels, which form the technical foundation, followed by a detailed introduction of the training and inference scheme of the consistency learning paradigm.

## 4.1 DIFFUSION TRAJECTORIES FOR LABELS

Recall the proposed SCL predicts labels using data inputs and noised label hints and pursues the predictive consistency across different noise steps, as shown in Fig. 1. This section elucidates the diffusion processes designed to gradually incorporate noise into labels across various label spaces.

**Diffusion on Categorical Labels.** For multi-dimensional categorical labels in $\{1, \cdots, K\}^N$ where $K$ denotes the category number and $N$ denotes the dimension (which could correspond to nodes in a graph and pixels in an image), we follow discrete diffusion models (Sohl-Dickstein et al., 2015; Austin et al., 2021; Hoogeboom et al., 2021) to model the diffusion process as introducing multinomial noise to the label at each timestep. We represent the label as $\mathbf{y} \in \{0, 1\}^{N \times K}$, which is a concatenation of $N$ one-hot vectors, each representing the categorical assignment of the corresponding dimension. At each timestep $t$, noise is applied to corrupt the one-hot representation of the label. This noise can be understood as transitioning between different categories for each of the $N$ dimensions. Specifically, starting from the initial point $\mathbf{y}_0 = \mathbf{y}$, the forward diffusion process is defined as:

$$q(\mathbf{y}_t|\mathbf{y}_{t-1}) = \text{Cat}(\mathbf{y}_t; \mathbf{p} = \mathbf{y}_{t-1}\mathbf{Q}_t), \tag{1}$$

where $\text{Cat}(\mathbf{y}; \mathbf{p})$ is categorical distributions over $N$ one-hot vectors with probabilities given by $\mathbf{p}$, and $\mathbf{Q}_t = (1 - \beta_t)\mathbf{I} + \beta_t/K\mathbf{1}\mathbf{1}^\top \in \mathbb{R}^{K \times K}$ is the transition matrix, which determines the corruption introduced at timestep $t$, where $\beta_t$ is the corruption rate at timestep $t$. This ensures that with probability $\beta_t$, the corresponding label category can transition to any other category, effectively introducing noise by redistributing the probability mass across categories. Over time, as $t$ increases, the labels become progressively noisier, eventually converging towards a uniform distribution over the $K$ categories (Austin et al., 2021). The cumulative effect of the diffusion process after $t$ steps is:

$$q(\mathbf{y}_t|\mathbf{y}_0) = \text{Cat}(\mathbf{y}_t; \mathbf{p} = \mathbf{y}_0\bar{\mathbf{Q}}_t), \tag{2}$$

where $\bar{\mathbf{Q}}_t = \mathbf{Q}_1\mathbf{Q}_2 \ldots \mathbf{Q}_t$ represents the accumulated transition matrix from $\mathbf{y}_0$ to $\mathbf{y}_t$.

**Diffusion on Continuous Labels.** For multi-dimensional continuous labels in $\mathbb{R}^N$, where $N$ denotes the dimensionality (which could correspond to regression outputs, pixel intensities in an image, or time-series data), we follow Gaussian diffusion models (Sohl-Dickstein et al., 2015; Ho et al., 2020; Nichol & Dhariwal, 2021) to model the diffusion process as introducing Gaussian noise to the label at each timestep. We represent the label as $\mathbf{y} \in \mathbb{R}^N$, where each element corresponds to a continuous value. At each timestep $t$, Gaussian noise is applied to corrupt the label, progressively pushing it toward a noisy distribution. Specifically, the forward diffusion process is defined as:

$$q(\mathbf{y}_t|\mathbf{y}_{t-1}) = \mathcal{N}(\mathbf{y}_t; \sqrt{1 - \beta_t}\mathbf{y}_{t-1}, \beta_t\mathbf{I}), \tag{3}$$

where $\mathcal{N}(\mathbf{y}; \mu, \Sigma)$ is a Gaussian distribution with mean $\mu$ and covariance $\Sigma$, and $\beta_t$ controls the variance of the added noise at timestep $t$. The factor $\sqrt{1 - \beta_t}$ ensures that the label retains some of its original value, while the noise is introduced with variance $\beta_t$, progressively corrupting the label as $t$ increases. Over time, as $t$ approaches the final timestep $T$, the labels become almost entirely corrupted, converging towards a Gaussian distribution centered at zero. The cumulative effect of this diffusion process after $t$ steps is described by the marginal distribution:

$$q(\mathbf{y}_t|\mathbf{y}_0) = \mathcal{N}(\mathbf{y}_t; \sqrt{\bar{\alpha}_t}\mathbf{y}_0, (1 - \bar{\alpha}_t)\mathbf{I}), \tag{4}$$

where $\bar{\alpha}_t = \prod_{i=1}^t (1 - \beta_i)$ represents the accumulated noise scale from the original label $\mathbf{y}_0$ to the noisy label $\mathbf{y}_t$. As $t$ increases, $\bar{\alpha}_t$ decreases, leading to increased corruption of the label.

## 4.2 Supervised Consistency Training Scheme

To better capture $I(X_E, Y)$, where $Y$ contains a substantial amount of information, directly maximizing the mutual information between $X_E$ and $Y$ can be challenging due to the sheer complexity and size of the label space. Instead of attempting to learn the entire information content of $Y$ at once, we aim to provide the model with a structured learning process that progressively captures this information. By exposing the model to noisy versions of $Y$, we can guide it to learn partial information at each step, using these noise-perturbed labels as hints for gradually reconstructing the full information content of $Y$. This progressive approach enables the model to focus on simpler aspects of the label information initially and incrementally build towards mastering the full complexity of $Y$. This gradual learning framework not only makes the task of capturing $I(X_E, Y)$ more tractable but also leverages the inherent structure and complexity of $Y$ to guide the progressive learning process.

This noise-based reconstruction process is similar to learning the consistency mappings in consistency models (Song et al., 2023; Song & Dhariwal, 2024), where the goal is to learn how to map back to the original data from different noise levels along the diffusion trajectories. In continuous-time diffusion models defined on $(\epsilon, T]$ (Song et al., 2020b), consistency models (Song et al., 2023) defines the self-consistency property as *points on the same trajectory map to the same initial point*, and optimize the learned consistency function $f_\theta(\cdot, \cdot)$ to satisfy the requirement by: 1) boundary condition: $f_\theta(\mathbf{y}_\epsilon, \epsilon) = \mathbf{y}_\epsilon$; 2) self-consistency property: $f_\theta$ outputs consistent estima-

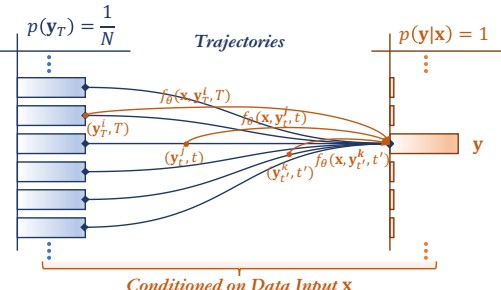

Figure 2: Prediction consistency enforces that all trajectories conditioned on $\mathbf{x}$ consistently map to the same initial point, i.e., the label $\mathbf{y}$.

tion for arbitrary pairs of $(\mathbf{y}_t, t)$ that belong to the same trajectory, i.e., $f_\theta(\mathbf{y}_t, t) = f_\theta(\mathbf{y}_{t'}, t'), \forall\, t, t' \in [\epsilon, T]$. The joint effect of these two constraints gradually transmits mapping consistency from low noise to high noise, and the model gradually learns how to restore the original data in the presence of higher information loss and finally achieve a reliable data prediction from noise step $T$ to data, i.e., $f_\theta(\mathbf{y}_T, T) \to \mathbf{y}_\epsilon$. This directly aligns with our goal of gradually learning $I(X_E, Y)$ in the supervised learning context, as discussed earlier. Yet, the difference from the raw consistency learning process is that the diffusion trajectories are conditioned on the data input $\mathbf{x}$ with a reference optimal solution $\mathbf{y}$ serving as the commonly targeted initial point for all the conditional trajectories. Thus, we define the consistency condition in the supervised learning context for model optimization below.

**Definition 4.1** (Prediction Consistency). Given data input $\mathbf{x}$ and a label trajectory $\{\mathbf{y}_t\}_{t \in [0,T]}$, we define the consistency function as $f : (\mathbf{x}, \mathbf{y}_t, t) \mapsto \mathbf{y}$, which satisfies: conditioned on $\mathbf{x}$, all points along any trajectory map to its label, i.e., $f_\theta(\mathbf{x}, \mathbf{y}_t^i, t) = f_\theta(\mathbf{x}, \mathbf{y}_{t'}^j, t') = \mathbf{y}$ for distinct trajectories $i$ and $j$ at distinct steps $t$ and $t'$.

As illustrated in Fig. 2, the goal of the consistency model $f_\theta$ in the supervised learning context is to estimate the consistency function from data by learning to enforce prediction consistency. To achieve such consistency to learn $f : \mathbf{x} \mapsto \mathbf{y}$, given that the target $\mathbf{y}$ is certain and explicit, we do not have to rely on optimizing the expectation of the variation of the consistency mappings over two noise points $\mathbf{y}_t$ and $\mathbf{y}_{t'}$ to propagate the label information across different noise levels. Instead, we additionally introduce $\mathbf{y}$ to optimize the triadic distance to achieve prediction consistency:

$$\mathcal{L}_{\text{PC}}(\theta) = \mathbb{E}\big[\lambda_1 \underline{d\big(f_\theta(\mathbf{x}, \mathbf{y}_t, t), \mathbf{y}\big)} + \lambda_1 \underline{d\big(f_\theta(\mathbf{x}, \mathbf{y}_{t'}, t'), \mathbf{y}\big)} + \lambda_2 \underline{d\big(f_\theta(\mathbf{x}, \mathbf{y}_t, t), f_\theta(\mathbf{x}, \mathbf{y}_{t'}, t')\big)}\big]. \quad (5)$$

Here $d(\cdot, \cdot)$ is a distance metric function and $\lambda_1, \lambda_2$ are loss weights. In this framework, the boundary conditions lose their critical importance since the information from $\mathbf{y}$ is gradually distributed across all noise stages. This allows the network $\theta$ to effectively model the consistency function $f_\theta$ over the entire progression. Thus, the core difference with the raw consistency model is that SCL aims to recover the exact $\mathbf{y}$ given $\mathbf{x}$, where the target distribution converges to an exact target point, and the model trades the output diversity to better capture $\mathbf{y}$. This calls for the requirement of consistency extending across all trajectories, rather than being confined within a single trajectory.

---

**Algorithm 1** Consistency Training Procedure

1: **Input:** Dataset $\mathcal{D}$, model $f_\theta$, noise function $q(\cdot)$, learning rate $\eta$, loss weights $\lambda_1, \lambda_2$
2: **repeat**
3:  Sample $(\mathbf{x}, \mathbf{y}) \sim \mathcal{D}$, and $t_1, t_2 \sim \text{Uniform}[1, T]$
4:  Sample $\mathbf{y}_{t_1} \sim q(\mathbf{y}_{t_1} | \mathbf{y}), \mathbf{y}_{t_2} \sim q(\mathbf{y}_{t_2} | \mathbf{y})$
5:  $\hat{\mathbf{y}}_0^{t_1} \leftarrow f_\theta(\mathbf{x}, \mathbf{y}_{t_1}, t_1)$
6:  $\hat{\mathbf{y}}_0^{t_2} \leftarrow f_\theta(\mathbf{x}, \mathbf{y}_{t_2}, t_2)$
7:  $\mathcal{L} \leftarrow \lambda_1 d(\hat{\mathbf{y}}_0^{t_1}, \mathbf{y}) + \lambda_1 d(\hat{\mathbf{y}}_0^{t_2}, \mathbf{y}) + \lambda_2 d(\hat{\mathbf{y}}_0^{t_1}, \hat{\mathbf{y}}_0^{t_2})$
8:  $\theta \leftarrow \theta - \eta \nabla_\theta \mathcal{L}$
9: **until** convergence

---

**Algorithm 2** Multistep Prediction

**Input:** trained model $f_\theta$, data input $\mathbf{x}$, noise function $q(\cdot)$, sequence of time points $\tau_1 > \tau_2 > \cdots > \tau_{N_\tau - 1}$

Sample random noise $\mathbf{y}_T$
$\hat{\mathbf{y}}_0 \leftarrow f_\theta(\mathbf{x}, \mathbf{y}_T, T)$
**for** $n = 1$ to $N_\tau - 1$ **do**
 Sample $\mathbf{y}_{\tau_n} \sim q(\mathbf{y}_{\tau_n} | \hat{\mathbf{y}}_0)$
 $\hat{\mathbf{y}}_0 \leftarrow f_\theta(\mathbf{x}, \mathbf{y}_{\tau_n}, \tau_n)$
**end for**
**Output:** Prediction $\hat{\mathbf{y}}_0$

---

Specifically, to align with the traditional supervised training paradigm, we retain the original task-defined loss function for the distance metric $d$, such as cross-entropy for classification tasks and mean squared error for regression tasks. This is because the design of the loss function is orthogonal to our learning framework, allowing them to complement each other. The main modification in our approach lies in that the model predicts $\mathbf{y}$ based on both $\mathbf{x}$ and the noise-perturbed versions of $\mathbf{y}$, while ensuring predictive consistency across different noise levels. In practice, with the noising schedules corresponding to different label spaces, we randomly sample two time steps $t_1$ and $t_2$ and independently apply noise to $\mathbf{y}$, and independently sample from the noise distribution to obtain $\mathbf{y}_{t_1}^i$ and $\mathbf{y}_{t_2}^j$. This ensures that the two noisy samples are independent with respect to both the time steps and the diffusion trajectories. Then Eq. 5 can be effectively optimized to learn the consistency predictive mapping, and the whole training process is presented in Alg. 1 and Fig. 1.

### 4.3 MULTISTEP INFERENCE WITH CONSISTENCY MAPPINGS

With a well-trained $f_\theta(\cdot, \cdot, \cdot)$, we obtain predictions for a given $\mathbf{x}$ by sampling $\mathbf{y}_T$ from the uniform distribution and then evaluate the prediction for $\mathbf{y}_0 = f_\theta(\mathbf{x}, \mathbf{y}_T, T)$. This standard single-step inference requires only one forward pass through the model, offering a fast yet approximate solution akin to conventional supervised learning methods. On the other hand, accuracy tends to be higher when $t$ is small, as the label hints contain a richer amount of information. Our objective is to progressively transfer this high accuracy to larger values of $t$ through training, thereby enhancing overall model performance. In the ideal case that the consistency loss converges to zero, optimal results can be achieved in a single step, yet in practice, gradually decreasing $t$ from $T$ to $0$ can lead to accuracy improvements. To achieve such enhancements, a multistep inference strategy can be adopted, which iteratively alternates between denoising and reintroducing noise. This approach effectively trades off runtime for enhanced prediction quality, allowing the model to refine its outputs over multiple inference steps and leverage increasingly rich information embedded in earlier predictions.

Given a sequence of time points $\tau_1 > \tau_2 > \cdots > \tau_{N_\tau - 1}$, at each step $\tau_n$, the current prediction $\mathbf{y}_{\tau_{n-1}}$ is perturbed by a noise function to a noisier state $\mathbf{y}_{\tau_n}$. The noise level decreases with each step, meaning $\tau_n < \tau_{n-1}$. The model then denoises the corrupted label by applying $f_\theta(\mathbf{x}, \mathbf{y}_{\tau_n}, \tau_n)$, producing a refined prediction. This process is repeated over successive steps, where each newly refined label incorporates progressively more accurate information from the previous step. This enables the model to gradually recover the whole information of $\mathbf{y}$ by taking the perhaps approximated prediction as the label hints and leveraging the incrementally informative hints for the final prediction. The specific multistep prediction procedure is presented in Alg. 2 and is visualized in Fig. 3.

## 5 EXPERIMENTS

We test the proposed SCL framework on tasks involving complex labels from diverse domains, including semantic segmentation (high-dimensional categorical outputs in vision learning), N-body simulation (high-dimensional continuous outputs in graph learning), combinatorial optimization problem solving (high-dimensional constrained combinatorial outputs in graph learning), and next-token prediction (high-dimensional sequential outputs in language modeling).

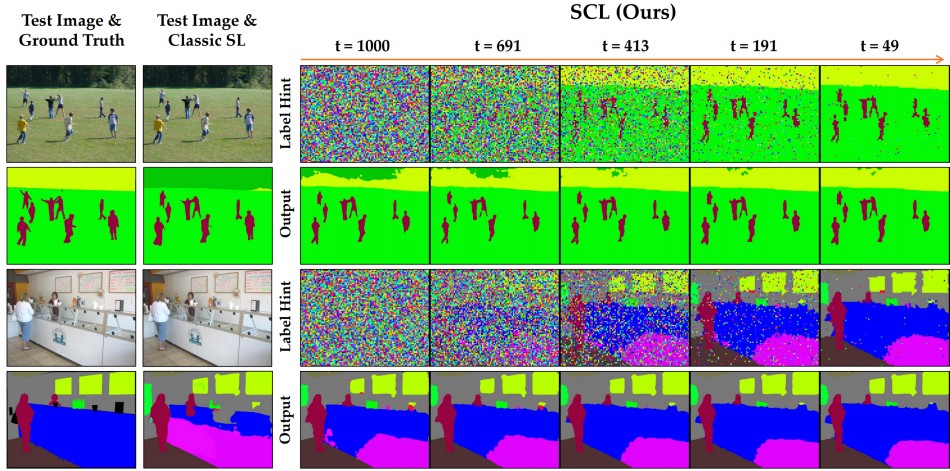

Figure 3: Predictions across varying timesteps based on the last step's predictions in the multistep inference procedure. In each step, the model receives the input and label hint and predicts the output.

Table 1: Results on Semantic Segmentation.

| Method | SCL | Pixel Acc.↑ | mIoU↑ | Score↑ |
|---|---|---|---|---|
| ResNet50dilated + PPM (Zhao et al., 2017) | ✗ | 77.64 | 39.58 | 58.61 |
| | ✓ | **78.63** | **39.70** | **59.17** |
| HRNetV2 (Sun et al., 2019) | ✗ | 78.71 | 40.77 | 59.74 |
| | ✓ | **79.54** | **42.68** | **61.11** |

## 5.1 SEMANTIC SEGMENTATION

Semantic segmentation is a classic dense vision task with wide applications that involves classifying each pixel of an image into a predefined category (Long et al., 2015; Zhao et al., 2017). Unlike classification tasks that categorize entire images, semantic segmentation analyzes the finer granularity of images to identify the boundaries and relationships between objects.

**Dataset.** We utilize the ADE20K dataset (Zhou et al., 2019), which is a commonly used large-scale scene parsing dataset that contains over 20,000 images with pixel-level annotations. The dataset is annotated with 150 different object classes, and we make the unannotated pixels into a new category, denoted as -1, which are ignored during both training and testing. Following previous works (Zhou et al., 2017; 2019), we resize the origial images during training while keeping the aspect ratio constant, randomly scaling the shorter side to one of the sizes 300, 375, 450, 525, or 600.

**Metrics.** Following (Zhou et al., 2017; 2019), we adopt three evaluation metrics to measure model performance: 1) Pixel Accuracy: the proportion of correctly classified pixels. 2) Mean IoU (mIoU): the intersection-over-union between the predicted and ground-truth pixels, averaged over all the classes. 3) Score: the average value of Pixel Accuracy and Mean IoU. During the testing phase, we use Multi-Scale Test: evaluate at multiple sizes and then take the average.

**Model Design.** We generally adopt an encoder-decoder network framework. The encoder compresses the input by extracting high-level features using a CNN backbone, reducing the spatial resolution while capturing important semantic information. The decoder then progressively upsamples the compressed features to recover the original resolution, often using skip connections to retain fine details. To introduce SCL, we concatenate the image features obtained through the encoder, the timestep embeddings extracted through sinusoidal position embedding, and the noised labels processed by the embedding layer. Then, we feed the tensor encompassing the input, timestep, and noised label into the decoder for further processing. In this task, $\mathbf{y} \in \{-1, 0, 1, ...149\}^{H \times W}$ where each entry will be converted into a one-hot vector of length 151, indicating the classification of the pixels. We adopt the categorical noising process as presented in Eq. 2 using transition matrices of $\mathbf{Q}_t \in [0, 1]^{151 \times 151}$.

**Results.** For the encoder, we choose ResNet50dilated (He et al., 2016), and HRNetV2 (Sun et al., 2019). For the decoder, we sequentially selected C1 (one convolution module) with DeepSup (deep supervision trick), PPM (Pyramid Pooling Module) (Zhao et al., 2017) with DeepSup, and C1. Table 1

Table 3: Comparison of traditional supervised learning and supervised consistency learning for MSE.

| Method | MSE↓ | MAE↓ |
|---|---|---|
| GCN (Kipf & Welling, 2016) | 0.01064±0.00014 | 0.04322±0.00082 |
| GCN-SCL (Ours) | **0.00927±0.00020** | **0.03783±0.00018** |
| GAT (Veličković et al., 2017) | 0.00969±0.00040 | 0.03996±0.00198 |
| GAT-SCL (Ours) | **0.00910±0.00038** | **0.03726±0.00068** |
| GGNN (Li et al., 2015) | 0.01220±0.00020 | 0.04614±0.00146 |
| GGNN-SCL (Ours) | **0.01143±0.00042** | **0.04336±0.00127** |

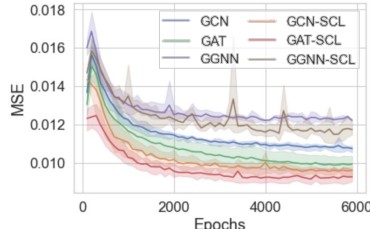

Figure 4: MSE curves on test data.

shows that SCL with merely one step achieved 2.63% performance gain in average pixel accuracy and 2.19% performance gain on mIoU. Fig. 3 visually demonstrates how increasing inference steps further improves predictions, particularly for large background areas.

## 5.2 N-BODY SIMULATION

The N-body simulation task involves predicting the future positions of a set of interacting particles over time based on initial conditions such as their positions, velocities, and the inherent physical forces governing their interactions (Satorras et al., 2021). The task's outputs retain the same dimensionality and complexity as the inputs. The evolution of particle positions and velocities follows fundamental physical

Table 2: Ablation study on loss construction.

| Method | MSE↓ | MAE↓ |
|---|---|---|
| Traditional SL | 0.01064 | 0.04322 |
| w/o $\lambda_1$-term, w/o $\lambda_2$-term | 4.36176 | 1.62478 |
| w/o $\lambda_1$-term, w/ $\lambda_2$-term | 4.34559 | 1.62437 |
| w/ $\lambda_1$-term, w/o $\lambda_2$-term | 0.00956 | 0.03895 |
| w/ $\lambda_1$-term, w/ $\lambda_2$-term | 0.00927 | 0.03783 |

laws such as gravitational or electrostatic interactions. We follow Satorras et al. (2021) to solve the 5-charged-particle system in 3-dimensional space. The system consists of five particles, each with either a positive or negative charge, and each particle has an associated position and velocity.

**Dataset.** We collected 3000 trajectories for training, 2000 for validation, and 2000 for testing. Each trajectory spans 1000 timesteps. For each trajectory, the initial conditions include the particle positions $p(0) = \{p_1(0), \ldots, p_5(0)\} \in \mathbb{R}^{5\times3}$, the initial velocities $v(0) = \{v_1(0), \ldots, v_5(0)\} \in \mathbb{R}^{5\times3}$, and the respective charges $c = \{c_1, \ldots, c_5\} \in \{-1, 1\}^5$. The task is to predict the positions of the five particles after 1000 timesteps. The model is optimized by minimizing the averaged Mean Squared Error (MSE) between the predicted positions and the ground truth positions.

**Metrics.** We adopt two evaluation metrics to evaluate the regression quality for test data: 1) Mean Square Error (MSE): the average of the squares of the errors between the predicted values and the true values; 2) Mean Absolute Error (MAE): the average of the absolute differences.

**Model Design.** We consider the state-of-the-art graph modeling solution for this task, where we input the concatenation of the initial positions and the velocities as the node features. The charges are input as edge attributes $a_{ij} = c_i c_j$. We take the model outputs as the estimated positions. To introduce SCL, we adopt two linear layers to encode the input attributes and the noised label, respectively, and then concatenate them to form the input hidden feature to the subsequent graph neural layers. We adopt 4 graph neural layers, and for each layer's output, we integrate the timestep embedding extracted by the sinusoidal position embedding and a linear layer through addition. In this task, $\mathbf{y} \in \mathbb{R}^{5\times3}$ and we adopt the Gaussian noising process to produce noised labels as shown in Eq. 4.

**Results.** We compare the model with the classic graph neural networks, including Graph Convolutional Networks (GCN) (Kipf & Welling, 2016), Graph Attention Network (GAT) (Veličković et al., 2017), and Gated Graph Neural Networks (GGNN) (Li et al., 2015). For each model, we compare the performance with the models trained by the classic SL and our proposed SCL. Table 3 shows the superiority of SCL on quantitative results with lower estimation errors on both MSE and MAE under same settings, and Fig. 4 shows performance gain on the test MSE curves within the training process. Table 2 provides ablation studies on the effects of the $\lambda_1$-term and $\lambda_2$-term and verifies the effectiveness of every loss term in Eq. 5.

## 5.3 COMBINATORIAL OPTIMIZATION

Combinatorial Optimization (CO) problems, which involve optimizing discrete variables under given objectives, generally maintain inherent computational difficulty, e.g. NP-hardness. Adopting the conventions established in Wang et al. (2022a), we define a CO problem on a graph $G(V, E)$, where

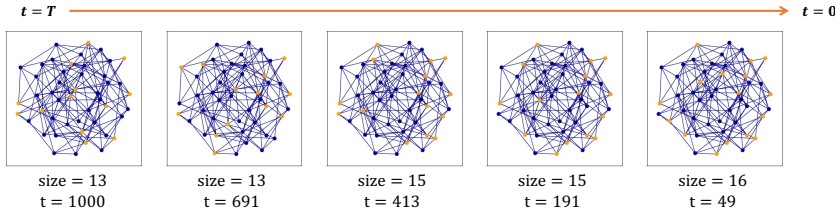

|  |  |  |  |  |
|---|---|---|---|---|
| size = 13 | size = 13 | size = 15 | size = 15 | size = 16 |
| t = 1000 | t = 691 | t = 413 | t = 191 | t = 49 |

Figure 5: Prediction results for the MIS solution based on the last step's predictions across varying time steps in the multistep inference procedure. Orange indicates the solution.

Table 4: Results on MIS. TS: Tree Search, UL: Unsupervised Learning. * denotes results quoted from previous works (Li et al., 2023; Zhang et al., 2023).

| Method | Type | RB-[200-300] | | | ER-[700-800] | | |
|---|---|---|---|---|---|---|---|
| | | Size↑ | Drop↓ | Time | Size↑ | Drop↓ | Time |
| Intel (Li et al., 2018) | SL+G | – | – | – | 34.86 | 22.31% | 6.06m |
| DIMES (Qiu et al., 2022) | RL+G | – | – | – | 38.24 | 14.78% | 6.12m |
| DIFUSCO (Sun & Yang, 2023) | SL+G | 18.52 | 7.81% | 16m3s | 37.03 | 18.53% | 5m30s |
| GCN (Kipf & Welling, 2016) | SL+G | 18.22 | 9.23% | 23s | 35.35 | 21.22% | 12s |
| GCN-SCL (T$_s$=1) | SL+G | 18.59 | 7.37% | **35s** | 36.72 | 18.17% | **11s** |
| GCN-SCL (T$_s$=5) | SL+G | **18.74** | **6.65%** | 1m16s | **37.80** | **15.76%** | 24s |
| Intel (Li et al., 2018) | SL+TS | 18.47 | 8.11% | 13m4s | 38.80 | 13.43% | 20.00m |
| DGL (Böther et al., 2022) | SL+TS | 17.36 | 13.61% | 12m47s | 37.26 | 16.96% | 22.71m |
| DIFUSCO (Sun & Yang, 2023) | SL+S | 19.13 | 4.79% | 20m28s | 39.12 | 12.81% | 21m43s |
| GCN (Kipf & Welling, 2016) | SL+S | 18.22 | 9.23% | 26s | 35.35 | 21.22% | 25s |
| GCN-SCL (T$_s$=1) | SL+S | 18.91 | 5.81% | **42s** | 37.91 | 15.52% | **24s** |
| GCN-SCL (T$_s$=5) | SL+S | **19.38** | **3.46%** | 1m50s | **39.81** | **11.27%** | 1m16s |

$V$ and $E$ denote the nodes and edges, respectively. Let $\mathbf{x} \in \{0,1\}^{N \times 2}$ denote the optimization variable, where each entry is represented by a one-hot vector, i.e., each entry with $(0,1)$ indicates that it is included in $\mathbf{x}$ and $(1,0)$ indicates the opposite. For node-decision problems, $\mathbf{x}_i$ indicates whether $V_i$ is included in $\mathbf{x}$. The feasible set $\Omega$ consists of $\mathbf{x}$ satisfying specific constraints as feasible solutions. A CO problem on $G$ aims to find a feasible $\mathbf{x}$ that minimizes the given objective function $l(\cdot; G)$. Here, we consider the classic node-decision problem Maximal Independent Set (MIS). Given an undirected graph $G = (V, E)$, an independent set is a subset of vertices $S \subseteq V$ such that no two vertices in $S$ are adjacent in $G$. MIS entails finding an independent set of maximum cardinality in $G$.

**Datasets.** Two datasets are tested for the MIS problem including RB graphs (Zhang et al., 2023) and ErdsRnyi (ER) graphs (Erdős et al., 1960). We randomly sample 200 to 300 vertices uniformly and generate the graph instances. ER graphs are randomly generated with each edge maintaining a fixed probability of being present or absent, independently of the other edges. We adopt ER graphs of 700 to 800 nodes with the pairwise connection probability set as 0.15.

Table 5: Numerical results of the enhancements achieved by increasing denoising steps.

| #Steps | Greedy | | | Sampling | | |
|---|---|---|---|---|---|---|
| | Size↑ | Drop↓ | Time | Size↑ | Drop↓ | Time |
| 1 | 18.586 | 7.486% | 13s | 18.888 | 5.983% | 20s |
| 2 | 18.604 | 7.397% | 20s | 19.246 | 4.201% | 30s |
| 4 | 18.702 | 6.909% | 33s | 19.366 | 3.604% | 49s |
| 8 | 19.026 | 5.296% | 1m0s | 19.616 | 2.359% | 1m27s |
| 16 | 19.284 | 4.012% | 1m47s | 19.702 | 1.932% | 2m43s |
| 32 | 19.438 | 3.245% | 3m31s | 19.790 | 1.493% | 5m14s |

**Metrics.** Following previous works (Kool et al., 2018; Joshi et al., 2019), we adopt three evaluation metrics to measure model performance: 1) Size: the average size of the solutions w.r.t. the corresponding instances, i.e. the objective. 2) Drop: the performance drop w.r.t. size compared to the optimal solution; 3) Time: the average computational time required to solve the problems.

**Model Design.** We primarily include graph networks to receive the graph input and output a binary prediction for each node, indicating the probability of the node being included in the optimal solution. To introduce SCL, the graph network receives the noised label and the adjacency matrix for the input node feature and edge feature, respectively. We adopt 12 graph neural layers, and for each layer's output, we integrate the timestep embedding extracted by the sinusoidal position embedding and a linear layer through addition. In this task, $\mathbf{y} \in \{0,1\}^{n \times 2}$ where each entry is a one-hot vector indicating the selection of the node. We adopt the categorical noising process as presented in Eq. 2.

**Results.** We compare GCN (Kipf & Welling, 2016) under the classic SL and SCL settings, and we also include other mainstream neural solvers into comparison with greedy and sampling decoding

($\times 4$). $T_s$ indicates the number of inference steps during the multistep prediction. Table. 4 shows the superiority of SCL on solving performance reflected by the solved size and relative drop. We also show that more inference steps can effectively improve the results in this scenario. Fig. 5 shows the result variation across the multistep prediction procedure, which gradually improves the prediction. Table 5 shows the numerical results variation across the multistep prediction procedure.

## 5.4 NEXT-TOKEN PREDICTION

The next-token prediction task is a cornerstone in natural language processing, forming the foundation for transformer-based large language models (LLMs) such as GPT (Radford, 2018), LLaMA (Touvron et al., 2023). The objective of the task is to predict the next token in a sequence, given the preceding context. The task's outputs retain similar dimensionality and sequence complexity as the inputs. Large language models trained on next-token prediction tasks have proven to be highly effective at capturing both short-term and long-range dependencies in language, enabling them to generate coherent, contextually appropriate text. This section investigates the effectiveness of SCL in the full fine-tuning task on the pre-trained LLaMa-2-7B (Touvron et al., 2023) models.

**Dataset.** The Alpaca (Taori et al., 2023) dataset is based on the self-instruct method (Wang et al., 2022b), utilizing the OpenAI text-davinci-003 engine to generate a collection of instructions and demonstrations. These instruction data can be employed for fine-tuning language models. By filtering out low-quality data, such as hallucinations, incorrect answers, and unclear instructions, we obtain the Alpaca-cleaned dataset, which serves as the sole training data for all models discussed in this paper.

**Metrics.** To evaluate the performance of models, we employ INSTRUCTEVAL (Chia et al., 2023), a comprehensive evaluation suite designed specifically for instruction-tuned models. INSTRUCTEVAL aims to assess models across dimensions such as problem-solving ability, writing proficiency, and alignment with human values. Following

Table 6: Evaluation on LLM fine-tuning.

| Method | MMLU↑ | CRASS↑ | BBH↑ |
|---|---|---|---|
| w/o FT | 41.90 | 37.59 | 32.93 |
| FT | 46.22 | 58.29 | 33.38 |
| FT-SCL (Ours) | **47.10** | **59.48** | **34.75** |

INSTRUCTEVAL, we use 5-shot direct prompting for MMLU (Hendrycks et al., 2020), 0-shot direct prompting for BBH (Srivastava et al., 2022) and 3-shot direct prompting for CRASS (Frohberg & Binder, 2022). Detailed descriptions for the benchmarks are presented in Appendix A.5.2.

**Model Design.** Our modified LLaMA2-7B model retains the original embedding and decoder layers but introduces a novel mechanism to predict the next token. Instead of the standard approach where hidden states directly generate token probabilities, we inject Gaussian noise into token embeddings during training to enhance robustness. This noise is combined with the tokens hidden state to form an augmented vector, which is passed through an MLP and classification head for prediction. Additionally, we enforce consistency by minimizing the mean squared error between logits at randomly selected time steps. In the generation phase, noise is iteratively reduced across steps to generate tokens sequentially. More detailed descriptions can be found in Appendix A.5.1.

**Results.** We compare methods for fine-tuning LLMs on the next-token prediction task. The baselines include the pre-trained LLaMA2-7B model and the LLaMA2-7B model fine-tuned using traditional full-parameter supervised learning (Taori et al., 2023). In contrast, our method, FT-SCL, also tunes the full parameters of the model but operates under the supervised consistency learning paradigm. We also include LoRA ($r = 8$, $\alpha = 16$, drop_out = 0.05) (Hu et al., 2021) for comparison. Table 6 shows the superiority of SCL on quantitative results across MMLU, GRASS and BBH.

## 6 CONCLUSION AND FUTURE WORK

This paper has proposed a novel supervised consistency learning framework beyond the classic supervised learning paradigm, which has shown superiority in extensive experiments on various tasks with complex labels. Our approach resembles, to a certain degree, the residual learning scheme by treating the noisy label as the input, which is counterpart to the raw signal of the input data. By leveraging label hints perturbed by noise and progressively refining predictions through multiple inference steps, our method improves predictive performance in challenging scenarios where traditional direct-label prediction methods may struggle. Empirical results across vision, text, and graph modalities demonstrate the superiority of the proposed paradigm. Future work will explore the potential of applying the data diffusion process to other deep learning paradigms, such as semi-supervised and unsupervised learning.

ETHICS STATEMENT

This research adheres to ethical standards and does not involve any direct human subjects, nor does it present any privacy or security concerns. The datasets used in this study are either synthetic or publicly available without involving sensitive or personally identifiable information. All experiments and methodologies were conducted in compliance with legal regulations and established research integrity practices. There are no known conflicts of interest, sponsorship influences, or concerns related to discrimination, bias, or fairness in our approach. Additionally, the research does not produce any harmful insights or applications.

REPRODUCIBILITY STATEMENT

We have taken steps to ensure the reproducibility of the results presented in this paper. The experimental settings, including datasets and model designs, are thoroughly described in Section 5. Additional details, such as model architectures, noise generation processes, and hyperparameter configurations, are provided in Appendix A. Source code will be made publicly available upon acceptance.

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

APPENDIX

# A  EXPERIMENT DETAILS

## A.1  COMPUTATIONAL RESOURCES.

Experiments for semantic segmentation, n-body simulation, and combinatorial optimization are conducted on a single GPU of NVIDIA RTX 4090, and experiments for next-token prediction are performed on 8 GPUs of NVIDIA H800.

## A.2  EXPERIMENTAL DETAILS FOR SEMANTIC SEGMENTATION

### A.2.1  DATASET

To accommodate the dimensions of the images output by the encoder, we downsample the segmentation maps. Additionally, to prevent rounding errors in subsequent calculations, it is necessary to adjust the target size's length and width according to an integer multiple of a constant determined by the padding parameter. Specifically, the downsampling rates for MobileNetV2dilated, ResNet50dilated, and HRNetV2 are 8, 8, and 4, respectively, with padding parameters of 8, 8, and 32 for these three encoders.

### A.2.2  CONTEXT AND TIME STEP

Given image information $\mathbf{x}$ extracted through the encoder, hint label $\mathbf{y}$, and time step $t$, SCL first embeds the latter two and then merges these three into a new image information $\tilde{\mathbf{x}}$. The context for semantic segmentation is generated by adding categorical noise to the ground truth labels. Given the context $y$, it is first passed through an embedding layer that maps each class $\mathbf{y} \in \{0, 1, \ldots, C\}$ (where $C$ is the number of classes) to a higher-dimensional vector $\tilde{\mathbf{y}}$. $\tilde{\mathbf{y}}$ is then processed through linear layers and activation function.

Time step $t$ is first embedded through the sinusoidal position embedding and then processed through linear layers and activation function.

$$\tilde{\mathbf{t}} = \text{concat}\left(\sin\frac{t}{T^{\frac{0}{d}}}, \cos\frac{t}{T^{\frac{0}{d}}}, \sin\frac{t}{T^{\frac{2}{d}}}, \cos\frac{t}{T^{\frac{2}{d}}}, \ldots, \sin\frac{t}{T^{\frac{d}{d}}}, \cos\frac{t}{T^{\frac{d}{d}}}\right) \tag{6}$$

$$\tilde{\mathbf{x}} = \text{concat}(x, W_2(\text{SiLU}(W_1\tilde{\mathbf{y}})), W_4(\text{SiLU}(W_3\tilde{\mathbf{t}}))) \tag{7}$$

where $d$ is the embedding dimension, $T$ is a large number (usually selected as 10000), $\text{concat}(\cdot)$ denotes concatenation.

## A.3  EXPERIMENTAL DETAILS FOR N-BODY SIMULATION

### A.3.1  NOISING PROCESS

At timestep $t = 0$, the label $\mathbf{y} \in \mathbb{R}^{5 \times 3}$ represents the original 3-dimensional coordinates of the 5-body system. We introduce Gaussian noise that gradually transforms the coordinates to points from the standard Gaussian distributions. The noising process simply follows Eq. 3 and Eq. 4.

### A.3.2  MODEL ARCHITECTURE

We follow the implementation of Satorras et al. (2021) to generally implement 4-layer GNNs. With its learnable edge operation function $\phi_e$ and node operation function $\phi_h$, the graph convolutional layer follows:

$$\mathbf{m}_{ij} = \phi_e(\mathbf{h}_i^l, \mathbf{h}_j^l, a_{ij}) \tag{8}$$

$$\mathbf{m}_i = \sum_{j \in \mathcal{N}(i)} \mathbf{m}_{ij} \tag{9}$$

$$\mathbf{h}_i^{l+1} = \phi_h(\mathbf{h}_i^l, \mathbf{m}_i) \tag{10}$$

Where $\mathbf{h}_i^l \in \mathbb{R}^{\text{nf}}$ is the nf-dimensional embedding of node $v_i$ at layer $l$. $a_{ij}$ are the edge attributes. $\mathcal{N}(i)$ represents the set of neighbors of node $v_i$. Here, $\phi_e$ and $\phi_h$ are approximated by 2-layer Multilayer Perceptrons (MLPs).

The initial position $\mathbf{p}^0$ and velocity $\mathbf{v}^0$ from the particles are passed through a linear layer to obtain the input feature. The label hint is passed through another linear layer, and the obtained feature is concatenated with the input feature and inputted into the GNN first layer $\mathbf{h}^0$. The particle's charges are inputted as edge attributes $a_{ij} = c_i c_j$. The time step $t$ is first embedded through the sinusoidal position embedding

$$\tilde{\mathbf{t}} = \text{concat}\left(\sin\frac{t}{T^{\frac{0}{d}}}, \cos\frac{t}{T^{\frac{0}{d}}}, \sin\frac{t}{T^{\frac{2}{d}}}, \cos\frac{t}{T^{\frac{2}{d}}}, \ldots, \sin\frac{t}{T^{\frac{d}{d}}}, \cos\frac{t}{T^{\frac{d}{d}}}\right) \tag{11}$$

and then processed through linear layers and activation functions. Here $d$ is the embedding dimension, $T$ is a large number (usually selected as 10000), $\text{concat}(\cdot)$ denotes concatenation. Then we aggregate the timestep feature with the node convolutional feature and reformulate the update for node features, i.e., Eq. 10 as:

$$\mathbf{h}_i^{l+1} = \phi_h(\mathbf{h}_i^l, \mathbf{m}_i) + \phi_t(\tilde{\mathbf{t}}) \tag{12}$$

where $\phi_h$ is a linear layer model. The output of the GNN $\mathbf{h}^L$ is passed through a two layers MLP that maps it to the estimated position.

## A.4 Experimental Details for Combinatorial Optimization

### A.4.1 Label Encoding and Noising Process

We represent the solutions of CO problems as $\mathbf{x} \in \{0,1\}^{N \times 2}$ with $\mathbf{x} \in \Omega$. The distribution of $\mathbf{x}$ is represented by $N$ Bernoulli distributions indicating whether each entry should be selected, i.e., $p(\mathbf{x}) \in [0,1]^{N \times 2}$. We try to establish transition trajectories from random uniform noise to high-quality soft-constrained solutions, i.e., $\mathbf{x} \in \{0,1\}^{N \times 2}$. These soft-constrained solutions are directly sampled from the estimated Bernoulli distributions, where feasibility constraints can be broadly captured through learning and eventually hard-guaranteed by post-processing.

The noising process is formulated as $q(\mathbf{x}_{1:T}|\mathbf{x}_0) = \prod_{t=1}^T q(\mathbf{x}_t|\mathbf{x}_{t-1})$, which is achieved by multiplying $\mathbf{x}_t \in \{0,1\}^{N \times 2}$ at step $t$ with a forward transition probability matrix $\mathbf{Q}_t \in [0,1]^{2 \times 2}$ which indicates the transforming probability of decision state. We set $\mathbf{Q}_t = \begin{bmatrix} \beta_t & 1-\beta_t \\ 1-\beta_t & \beta_t \end{bmatrix}$ (Austin et al., 2021), where $\beta_t \in [0,1]$ such that the transition matrix is doubly stochastic with strictly positive entries, ensuring that the stationary distribution is uniform which is an unbiased prior for sampling. The noising process for each step and the $t$-step marginal are formulated as:

$$q(\mathbf{x}_t|\mathbf{x}_{t-1}) = \text{Cat}(\mathbf{x}_t; \mathbf{p} = \mathbf{x}_{t-1}\mathbf{Q}_t) \quad \text{and} \quad q(\mathbf{x}_t|\mathbf{x}_0) = \text{Cat}(\mathbf{x}_t; \mathbf{p} = \mathbf{x}_0\overline{\mathbf{Q}}_t) \tag{13}$$

where $\text{Cat}(\mathbf{x}; \mathbf{p})$ is a categorical distribution over $N$ one-hot variables and $\overline{\mathbf{Q}}_t = \mathbf{Q}_1\mathbf{Q}_2\cdots\mathbf{Q}_t$.

### A.4.2 Model Architecture

The MIS problem can be defined as $G(V, E)$, where $V \in \{0,1\}^N$ and $E \in \mathbb{N}^{N \times 2}$ denote the nodes and edges, respectively. For the classic supervised learning, $V$ is a fully one vector, while for SCL $V$ is an indicator zero-one vector derived from the ground truth with noise added at time step $t$. $E$ is the edge index, which contains all pairs of connected nodes.

**Input Embedding Layer.** For node vector $\mathbf{x} \in \{0,1\}^N$, each node will be mapped to a feature vector $\tilde{\mathbf{x}}$ of dimension $H$. For the time step $t$, it is embedded through the sinusoidal position embedding.

$$\tilde{\mathbf{t}} = \text{concat}\left(\sin\frac{t}{T^{\frac{0}{d}}}, \cos\frac{t}{T^{\frac{0}{d}}}, \sin\frac{t}{T^{\frac{2}{d}}}, \cos\frac{t}{T^{\frac{2}{d}}}, \ldots, \sin\frac{t}{T^{\frac{d}{d}}}, \cos\frac{t}{T^{\frac{d}{d}}}\right) \tag{14}$$

where $d$ is the embedding dimension, $T$ is a large number (usually selected as 10000), $\text{concat}(\cdot)$ denotes concatenation.

Next, we compute the input features of the graph convolution layer:

$$\mathbf{x}_i^0 = W_1 \tilde{\mathbf{x}}_i \tag{15}$$

$$\mathbf{t}^0 = W_3(\text{ReLU}(W_2 \tilde{\mathbf{t}})) \tag{16}$$

where $\mathbf{t}^0 \in \mathbb{R}^{d_t}$, $d_t$ is the time feature embedding dimension. Moreover, we initialize edge feature $\mathbf{e}^0$ to a zero matrix $\mathbf{0}^{E \times d}$.

**Graph Convolution Layer.** Following Joshi et al. (2019), the cross-layer convolution operation is formulated as:

$$\mathbf{x}_i^{l+1} = \mathbf{x}_i^l + \text{ReLU}(\text{BN}(W_1^l \mathbf{x}_i^l + \sum_{j \sim i} \eta_{ij}^l \odot W_2^l \mathbf{x}_j^l)) \tag{17}$$

$$\mathbf{e}_{ij}^{l+1} = \mathbf{e}_{ij}^l + \text{ReLU}(\text{BN}(W_3^l \mathbf{e}_{ij}^l + W_4^l \mathbf{x}_i^l + W_5^l \mathbf{x}_j^l)) \tag{18}$$

$$\boldsymbol{\eta}_{ij}^l = \frac{\sigma(\mathbf{e}_{ij}^l)}{\sum_{j' \sim i} \sigma(\mathbf{e}_{ij'}^l) + \epsilon} \tag{19}$$

where $x_i^l$ and $e_{ij}^l$ denote the node feature vector and edge feature vector at layer $l$, $W_1, \cdots, W_5 \in \mathbb{R}^{h \times h}$ denote the model weights, $\eta_{ij}^l$ denotes the dense attention map. The convolution operation integrates the edge feature to accommodate the significance of edges in routing problems.

we aggregate the timestep feature with the node convolutional feature and reformulate the update for node features as follows:

$$\mathbf{x}_i^{l+1} = \mathbf{x}_i^l + \text{ReLU}(\text{BN}(W_1^l \mathbf{x}_i^l + \sum_{j \sim i} \boldsymbol{\eta}_{ij}^l \odot W_2^l \mathbf{x}_j^l)) + W_6^l(\text{ReLU}(\mathbf{t}^0)) \tag{20}$$

**Output Layer.** The prediction of the node heatmap in MIS is as follows:

$$\mathbf{x}_i = \text{Softmax}(\text{norm}(\text{ReLU}(W_n \mathbf{x}_i^L))) \tag{21}$$

where $L$ is the number of GCN layers and norm is layer normalization.

### A.5 EXPERIMENTAL DETAILS FOR NEXT-TOKEN PREDICTION

#### A.5.1 MODEL DESIGN

In our approach, we adopt the LLaMA2-7B model as the backbone, preserving the structure of the embedding and decoder layers. However, we modify the prediction mechanism for the next token using the hidden states. In a conventional decoder-only language model (LLM), the prediction of the next token $y$ is achieved by leveraging the preceding context, encoded in a high-dimensional hidden state $h$. This hidden state $h$ is then passed through a linear layer, typically referred to as the language modeling head (lm_head) or unembedding layer, to yield the probability distribution $P(y)$ over the next token.

In contrast, our model introduces a noise injection mechanism to perturb the token embeddings, aiming to enhance robustness and generalization.

**Training Phase.** During the training phase, when the model obtains the last hidden states for each token, instead of directly passing them through the lm_head to generate logits and compute the cross-entropy loss with the ground truth labels $\hat{\mathbf{y}}$, we transform these labels back into its corresponding embedding $\mathbf{y}_{\text{emb}}$. Then add Gaussian noise $\epsilon \sim \mathcal{N}(0, \sigma^2)$ to $\mathbf{y}_{\text{emb}}$, resulting in a perturbed embedding:

$$\mathbf{y}_t^{\text{noisy}} = \bar{\alpha}_t \mathbf{y}^{\text{emb}} + \bar{\beta}_t \epsilon. \tag{22}$$

To inform the model of the noise magnitude, we also perturb the time step $t$, obtaining a corresponding time embedding $\mathbf{t}_{\text{emb}}$. The noisy token embedding $\mathbf{y}_t^{\text{noisy}}$ is combined with $\mathbf{t}_{\text{emb}}$ to form a new noisy information vector,

$$\mathbf{h}_t^{\text{noisy}} = \mathbf{y}_t^{\text{noisy}} + \mathbf{t}^{\text{emb}}. \tag{23}$$

This noisy information is concatenated with the hidden state $\mathbf{h}$, resulting in the augmented vector

$$\mathbf{h}_t^{\text{aug}} = [\mathbf{h}; \mathbf{h}_t^{\text{noisy}}]. \tag{24}$$

Finally, $\mathbf{h}_{\text{aug}}$ is passed through a multi-layer perceptron (MLP) and a new classification head to generate the probability distribution for the next token:

$$p(\mathbf{y} \mid \mathbf{h}_t^{\text{aug}}) = \text{softmax}(\text{LM\_HEAD}((\text{MLP}(\mathbf{h}_{\text{aug}})))). \tag{25}$$

It is worth noting that for each batch, In addition to aligning $p(\mathbf{y} \mid \mathbf{h}_t^{\text{aug}})$ with the next token ground truth by minimizing the cross-entropy loss, we randomly generate two time steps, $t_1$ and $t_2$, and obtain logits $\text{logits}(\mathbf{y} \mid \mathbf{h}_{t_1}^{\text{aug}})$ and $\text{logits}(\mathbf{y} \mid \mathbf{h}_{t_2}^{\text{aug}})$. We then minimize the mean squared error (MSE) loss between them to ensure as much consistency as possible.

**Generation Phase** Now we describe the generation process. After the input passes through the decoder layers and obtains the last hidden states, we encounter a challenge during the inference phase since the next token $\hat{\mathbf{y}}$ is unknown. To address this, we input a complete Gaussian noise vector $\mathbf{h}_{1000}^{\text{noisy}}$ (i.e. $\epsilon$), which is denoised by model to generate the next token $\mathbf{y}_1$. This process is then iterated, with $\mathbf{y}_1$ serving as $\hat{\mathbf{y}}$ for the subsequent iteration. Following the steps outlined in Eq. 22, 23, 24, and 25, we generate $\mathbf{y}_2$, and so on, iteratively.

The noise addition time step for each iteration is predetermined as a hyperparameter. In our experiments, we employ a linearly decreasing schedule for the time steps. For instance, with a maximum noise step of 1000 and 5 iterations, the time steps $t$ are set as $[1000, 800, 600, 400, 200]$, ensuring a gradual reduction of noise over the course of iterations.

### A.5.2 EVALUATION METRICS

The evaluation of large language model performance in this paper includes benchmarks:

- MMLU (Hendrycks et al., 2020): This benchmark assesses models' world knowledge and reasoning abilities across 57 academic disciplines, including STEM, humanities, and social sciences. Questions range in difficulty from elementary to advanced professional levels, presented in a multiple-choice format. The evaluation primarily uses few-shot settings to test the models' generalization capabilities. In our experiments, a 5-shot direct prompting is utilized to evaluate the model's comprehensive performance across various dimensions.

- BBH (Srivastava et al., 2022): It is a subset of 23 challenging tasks from the BIG-Bench benchmark. It evaluates the ability of models to handle challenging reasoning and problem-solving tasks that go beyond simple language understanding. It includes complex scenarios such as navigation, logical deduction, and fallacy detection. For this work, we apply 0-shot direct prompting to measure the model's capability in dealing with unseen questions without additional contextual examples.

- CRASS (Frohberg & Binder, 2022): This benchmark is designed to evaluate the model's ability to handle complex relational reasoning tasks, specifically in the context of causal structures and relationships. It includes a variety of problems that test how well the model understands and predicts causal relationships between different entities or events. For this evaluation, we use 3-shot direct prompting to assess model reasoning.

## B SUPPLEMENTARY EXPERIMENTAL RESULTS

For semantic segmentation, to vividly illustrate the effects of SCL, we present a comparison of the Intersection over Union (IoU) metrics for all three models across the ADE20K dataset's 150 categories in Table 7. The results indicate that MobileNetV2dilated, ResNet50dilated, and HRNetV2 have achieved advantages in IoU on 70.00%, 58.67%, and 64.67% of the categories respectively after using SCL.

Fig. 6 illustrates the differences in predicting semantic segmentation maps among various models and training methods, visually reflecting the performance differences between the models as outlined in Table 1.

Table 7: Comparison of Intersection over Union (IoU) for classic supervised learning (SL) versus the proposed supervised consistency learning (SCL) in semantic segmentation across various neural backbones. Bold indicates better performance in that category. MoibleNet: MobileNetV2dilated, ResNet50: ResNet50dilated. Order: Ranked from top to bottom based on the probability of each category in the ADE20K dataset

| Object | MobileNet SL | MobileNet SCL | ResNet50 SL | ResNet50 SCL | HRNetV2 SL | HRNetV2 SCL | Object | MobileNet SL | MobileNet SCL | ResNet50 SL | ResNet50 SCL | HRNetV2 SL | HRNetV2 SCL |
|---|---|---|---|---|---|---|---|---|---|---|---|---|---|
| wall | 55.17 | 52.67 | 71.98 | 72.80 | 73.95 | 71.89 | building | 60.33 | 65.75 | 80.03 | 80.19 | 79.61 | 80.82 |
| sky | 79.30 | 87.51 | 93.24 | 93.10 | 93.31 | 93.81 | floor | 54.27 | 58.03 | 76.31 | 76.73 | 77.55 | 78.81 |
| tree | 50.65 | 58.62 | 69.36 | 71.51 | 70.95 | 72.35 | ceiling | 58.45 | 71.21 | 78.46 | 80.11 | 81.31 | 81.71 |
| road | 34.60 | 66.73 | 79.53 | 79.54 | 80.43 | 80.90 | bed | 28.84 | 31.57 | 82.95 | 83.52 | 85.49 | 86.75 |
| window | 42.03 | 46.20 | 56.26 | 56.57 | 58.96 | 58.09 | grass | 16.51 | 12.29 | 64.23 | 67.29 | 64.20 | 65.97 |
| cabinet | 20.82 | 27.94 | 54.28 | 54.48 | 56.17 | 59.32 | sidewalk | 38.18 | 31.78 | 60.81 | 58.59 | 63.67 | 64.15 |
| person | 51.72 | 57.58 | 74.25 | 70.87 | 77.81 | 77.85 | earth | 16.74 | 20.66 | 30.05 | 34.30 | 31.14 | 30.08 |
| door | 18.40 | 24.87 | 34.94 | 41.76 | 44.42 | 43.97 | table | 31.27 | 33.32 | 50.23 | 53.99 | 55.03 | 57.98 |
| mount | 9.72 | 13.55 | 53.75 | 55.40 | 53.10 | 57.95 | plant | 28.90 | 32.02 | 43.80 | 46.78 | 49.04 | 49.07 |
| curtain | 41.14 | 39.88 | 64.09 | 66.14 | 67.57 | 68.59 | chair | 33.72 | 37.83 | 51.92 | 51.91 | 53.37 | 55.18 |
| car | 68.78 | 71.61 | 79.95 | 79.97 | 80.24 | 81.10 | water | 23.51 | 21.73 | 51.42 | 56.13 | 46.04 | 52.65 |
| picture | 51.24 | 51.66 | 62.99 | 68.86 | 66.10 | 68.41 | sofa | 38.90 | 35.38 | 55.97 | 58.36 | 65.39 | 64.57 |
| shelf | 21.15 | 23.20 | 37.45 | 42.28 | 36.91 | 32.79 | house | 0.66 | 0.26 | 53.76 | 47.20 | 40.27 | 45.35 |
| sea | 4.21 | 13.54 | 54.66 | 39.60 | 50.72 | 38.60 | mirror | 16.26 | 21.99 | 48.81 | 58.45 | 57.74 | 57.76 |
| rug | 28.43 | 28.09 | 50.92 | 56.00 | 50.72 | 57.74 | field | 13.88 | 19.86 | 21.73 | 21.02 | 30.34 | 30.11 |
| armchair | 19.71 | 31.42 | 36.04 | 41.07 | 41.94 | 46.86 | seat | 7.84 | 17.84 | 47.73 | 54.29 | 49.31 | 50.30 |
| fence | 13.16 | 21.20 | 29.07 | 28.81 | 32.07 | 34.36 | desk | 16.55 | 17.81 | 42.92 | 45.37 | 42.54 | 49.21 |
| rock | 14.56 | 8.42 | 35.75 | 43.65 | 40.28 | 43.53 | press | 13.25 | 14.50 | 41.17 | 45.74 | 42.14 | 47.93 |
| lamp | 40.87 | 42.23 | 56.26 | 57.37 | 61.33 | 60.19 | bath | 30.84 | 35.84 | 67.65 | 66.95 | 70.14 | 70.34 |
| railing | 8.62 | 12.23 | 31.97 | 34.03 | 24.83 | 23.84 | cushion | 29.01 | 29.01 | 45.97 | 45.37 | 51.44 | 52.76 |
| base | 4.38 | 0.66 | 24.39 | 25.46 | 17.36 | 29.95 | box | 5.19 | 8.89 | 14.65 | 17.20 | 15.75 | 17.49 |
| column | 20.77 | 20.55 | 39.33 | 47.18 | 44.92 | 48.44 | signboard | 22.52 | 25.51 | 29.49 | 23.82 | 33.32 | 28.03 |
| chest | 21.75 | 32.50 | 32.36 | 34.30 | 38.58 | 37.99 | counter | 4.35 | 9.60 | 18.40 | 44.91 | 18.65 | 31.87 |
| sand | 3.82 | 0.53 | 29.26 | 40.35 | 25.49 | 34.61 | sink | 28.45 | 39.79 | 64.95 | 64.96 | 62.80 | 68.05 |
| skyscraper | 20.65 | 12.74 | 48.96 | 53.04 | 31.32 | 40.56 | hearth | 37.53 | 54.30 | 55.77 | 59.66 | 66.93 | 68.47 |
| refrigerator | 17.92 | 41.57 | 66.13 | 60.50 | 57.07 | 56.39 | grandstand | 21.20 | 26.21 | 38.36 | 41.64 | 39.68 | 48.53 |
| path | 6.39 | 15.45 | 18.51 | 22.61 | 20.92 | 23.19 | stairs | 13.02 | 22.45 | 17.93 | 41.29 | 25.29 | 37.92 |
| runway | 24.24 | 30.83 | 57.01 | 47.99 | 69.66 | 63.95 | case | 18.17 | 17.95 | 42.02 | 49.53 | 33.40 | 51.27 |
| pool table | 69.94 | 85.99 | 89.27 | 87.82 | 90.97 | 91.95 | pillow | 20.01 | 28.61 | 43.04 | 45.81 | 49.79 | 55.60 |
| screen door | 2.26 | 0.00 | 57.19 | 61.30 | 62.64 | 67.65 | stairway | 8.87 | 16.92 | 15.61 | 17.07 | 19.60 | 18.40 |
| river | 6.08 | 2.75 | 10.58 | 15.01 | 10.39 | 11.18 | bridge | 11.26 | 10.04 | 51.08 | 55.82 | 50.82 | 48.94 |
| bookcase | 0.38 | 6.81 | 36.90 | 40.99 | 37.45 | 40.35 | blind | 8.16 | 14.51 | 36.17 | 48.03 | 30.68 | 41.88 |
| coffee table | 31.60 | 36.12 | 48.86 | 49.64 | 55.00 | 56.51 | toilet | 37.47 | 60.28 | 83.18 | 83.18 | 82.60 | 83.54 |
| flower | 11.66 | 27.67 | 30.84 | 30.62 | 33.09 | 28.74 | book | 2.27 | 12.11 | 39.20 | 41.51 | 44.86 | 39.72 |
| hill | 2.86 | 4.43 | 4.72 | 13.30 | 5.26 | 9.36 | bench | 12.83 | 20.20 | 36.10 | 33.18 | 38.11 | 38.58 |
| countertop | 13.96 | 10.53 | 47.49 | 47.28 | 48.32 | 47.91 | stove | 25.40 | 49.79 | 67.87 | 46.98 | 62.05 | 64.29 |
| palm | 19.33 | 17.12 | 45.77 | 43.52 | 49.66 | 50.78 | kitchen island | 6.66 | 15.62 | 33.46 | 13.36 | 25.16 | 26.50 |
| computer | 31.65 | 30.84 | 51.94 | 54.03 | 54.27 | 55.70 | swivel chair | 18.94 | 19.04 | 51.25 | 27.42 | 42.83 | 34.40 |
| boat | 9.29 | 7.08 | 49.44 | 51.81 | 36.32 | 63.33 | bar | 6.19 | 15.51 | 16.96 | 45.78 | 28.46 | 36.18 |
| arcade machine | 3.71 | 9.95 | 38.70 | 3.79 | 25.19 | 23.22 | hovel | 1.55 | 0.50 | 9.10 | 28.57 | 13.78 | 29.86 |
| bus | 25.60 | 17.04 | 75.65 | 72.93 | 81.86 | 81.73 | towel | 15.58 | 6.55 | 51.74 | 50.39 | 48.06 | 49.66 |
| light | 29.91 | 26.28 | 47.97 | 36.87 | 51.36 | 42.78 | truck | 1.04 | 4.17 | 14.06 | 11.60 | 13.25 | 15.08 |
| tower | 13.24 | 1.01 | 35.98 | 0.06 | 29.73 | 21.27 | chandelier | 37.23 | 46.96 | 60.83 | 58.58 | 65.80 | 64.43 |
| awning | 4.74 | 5.26 | 19.51 | 12.40 | 19.56 | 10.91 | streetlight | 4.26 | 4.74 | 25.25 | 17.23 | 27.41 | 16.96 |
| booth | 10.46 | 24.09 | 38.24 | 40.57 | 45.94 | 51.60 | tv set | 33.50 | 37.66 | 56.26 | 60.61 | 52.94 | 57.63 |
| airplane | 22.25 | 32.62 | 49.22 | 34.95 | 49.59 | 52.51 | dirt track | 4.14 | 0.00 | 6.30 | 0.00 | 7.04 | 9.60 |
| apparel | 10.07 | 22.31 | 27.23 | 33.09 | 26.55 | 29.69 | pole | 9.08 | 10.10 | 17.04 | 6.61 | 20.83 | 14.53 |
| land | 0.27 | 0.30 | 0.50 | 0.37 | 5.37 | 0.53 | bannister | 1.23 | 0.32 | 10.81 | 6.69 | 8.69 | 12.03 |
| escalator | 1.11 | 0.00 | 16.69 | 5.82 | 13.74 | 17.55 | ottoman | 9.98 | 24.71 | 36.78 | 46.73 | 30.40 | 32.51 |
| bottle | 2.33 | 1.89 | 33.48 | 14.25 | 25.57 | 15.74 | buffet | 13.08 | 34.27 | 34.20 | 8.13 | 24.84 | 31.98 |
| poster | 8.78 | 2.90 | 14.71 | 10.07 | 20.31 | 24.07 | stage | 2.56 | 3.56 | 3.92 | 8.10 | 9.77 | 13.98 |
| van | 21.37 | 39.01 | 43.27 | 39.23 | 39.35 | 42.44 | ship | 1.22 | 0.00 | 16.89 | 0.00 | 14.58 | 11.73 |
| fountain | 1.04 | 0.62 | 6.85 | 64.03 | 4.19 | 57.98 | conveyor | 19.62 | 31.73 | 37.20 | 23.91 | 48.30 | 51.15 |
| canopy | 2.47 | 6.64 | 13.97 | 12.56 | 15.29 | 18.08 | washer | 34.79 | 17.57 | 58.88 | 52.47 | 68.04 | 64.49 |
| plaything | 3.97 | 6.40 | 15.94 | 19.29 | 9.85 | 12.22 | natatorium | 18.05 | 41.07 | 27.20 | 0.28 | 37.30 | 35.04 |
| stool | 15.65 | 27.78 | 28.11 | 29.88 | 35.75 | 36.69 | barrel | 0.00 | 0.00 | 14.37 | 0.00 | 8.06 | 10.63 |
| basket | 9.14 | 12.13 | 24.75 | 23.71 | 22.46 | 23.42 | waterfall | 12.33 | 5.92 | 46.36 | 13.38 | 49.73 | 38.72 |
| tent | 6.92 | 10.27 | 70.53 | 86.04 | 83.03 | 82.68 | bag | 1.18 | 2.40 | 6.32 | 12.33 | 3.91 | 7.21 |
| minibike | 27.88 | 30.32 | 44.75 | 61.82 | 43.75 | 63.65 | cradle | 29.83 | 33.51 | 75.56 | 77.68 | 76.12 | 72.63 |
| oven | 14.56 | 30.68 | 27.24 | 40.18 | 43.96 | 52.18 | ball | 25.16 | 21.53 | 26.40 | 30.73 | 25.50 | 33.23 |
| food | 19.20 | 20.55 | 43.99 | 34.42 | 47.36 | 50.25 | step | 3.02 | 0.10 | 1.18 | 0.75 | 11.75 | 4.44 |
| tank | 0.00 | 9.53 | 20.04 | 42.80 | 20.91 | 47.38 | brand | 14.60 | 22.55 | 22.16 | 25.29 | 19.88 | 28.83 |
| microwave | 11.87 | 15.51 | 32.47 | 35.50 | 36.97 | 37.15 | pot | 20.34 | 26.40 | 34.15 | 34.99 | 39.20 | 34.24 |
| animal | 19.14 | 30.30 | 48.58 | 52.32 | 52.12 | 49.17 | bicycle | 34.86 | 33.71 | 46.94 | 45.47 | 41.60 | 45.71 |
| lake | 0.14 | 0.00 | 37.96 | 0.00 | 27.51 | 5.27 | dishwasher | 11.56 | 40.37 | 55.70 | 56.69 | 60.57 | 58.20 |
| screen | 26.05 | 42.53 | 60.32 | 65.38 | 64.16 | 64.48 | blanket | 5.27 | 2.55 | 6.07 | 18.39 | 8.25 | 19.16 |
| sculpture | 2.80 | 2.14 | 13.78 | 4.03 | 25.08 | 19.24 | hood | 5.68 | 20.41 | 40.03 | 36.07 | 49.03 | 48.95 |
| sconce | 9.51 | 21.67 | 34.37 | 36.90 | 39.11 | 39.53 | vase | 11.92 | 14.10 | 32.35 | 35.54 | 29.51 | 28.97 |
| stoplight | 3.68 | 0.79 | 29.06 | 22.45 | 27.53 | 24.33 | tray | 0.66 | 7.20 | 5.23 | 0.84 | 7.16 | 3.78 |
| ashcan | 11.77 | 28.05 | 38.41 | 41.58 | 38.46 | 36.84 | fan | 29.23 | 34.55 | 50.62 | 39.84 | 56.09 | 50.77 |
| pier | 0.87 | 6.88 | 39.02 | 29.96 | 20.96 | 38.67 | crt screen | 3.50 | 5.05 | 1.84 | 5.49 | 8.34 | 9.01 |
| plate | 14.07 | 24.38 | 36.25 | 36.12 | 34.82 | 31.07 | monitor | 4.47 | 21.44 | 2.09 | 46.57 | 16.12 | 23.76 |
| notice board | 18.78 | 3.77 | 30.75 | 41.06 | 31.50 | 29.33 | shower | 0.00 | 0.00 | 0.17 | 0.00 | 0.17 | 0.00 |
| radiator | 13.50 | 19.97 | 39.72 | 43.61 | 40.29 | 42.46 | glass | 0.59 | 5.68 | 11.82 | 9.47 | 11.16 | 10.19 |
| clock | 3.96 | 8.35 | 26.99 | 20.58 | 19.74 | 15.70 | flag | 3.91 | 15.81 | 19.59 | 20.53 | 26.78 | 29.24 |

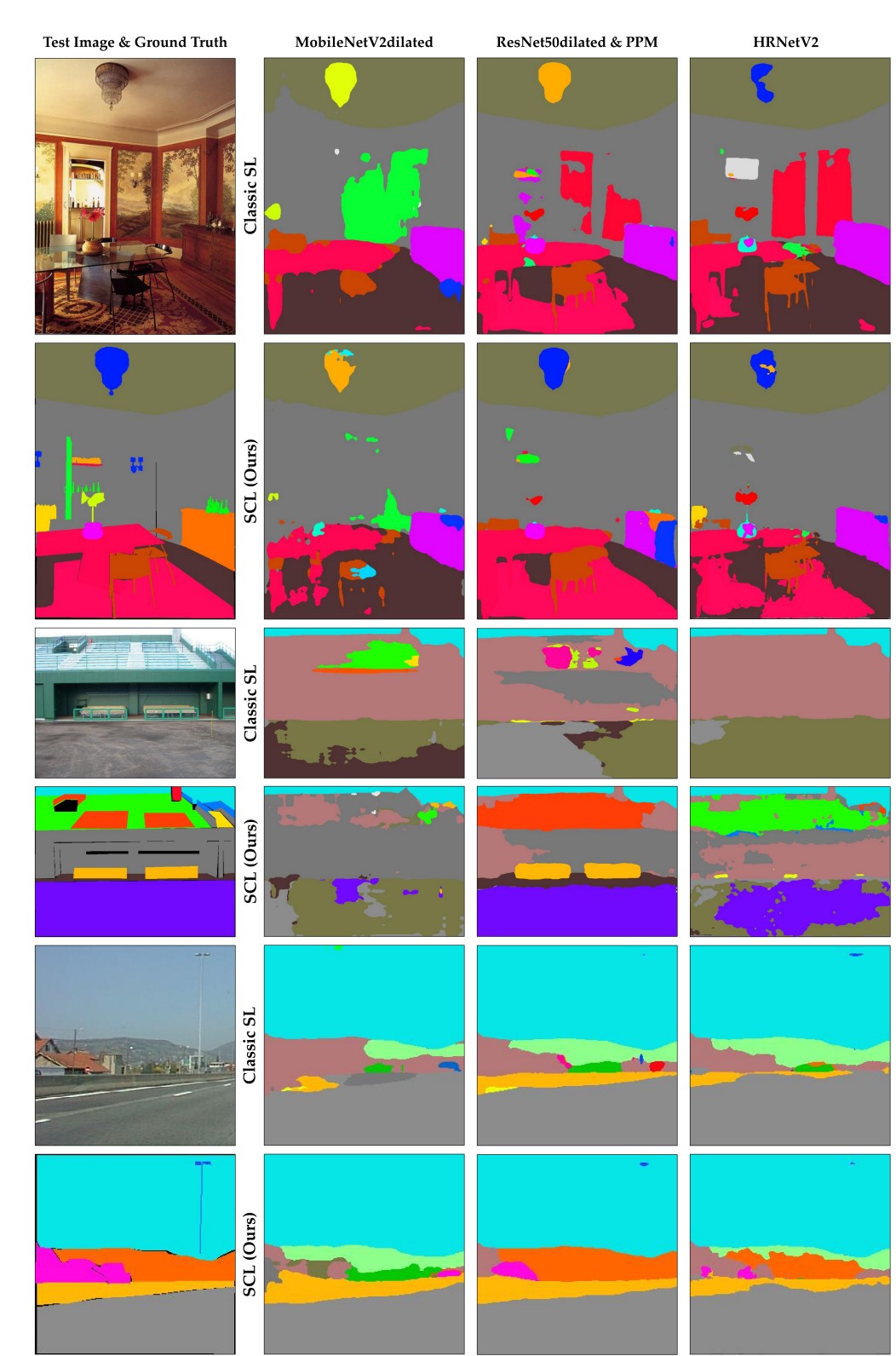

Figure 6: Comparison of classic supervised learning (SL) and the proposed supervised consistency learning (SCL) on semantic segmentation across different neural backbones.

