# OpenReview forum: "Your Consistency Model is Secretly a More Powerful Supervised Learning Paradigm for Learning Tasks with Complex Labels"
_ICLR.cc/2025/Conference — Submitted to ICLR 2025_

### Official Review · Reviewer_zvSo · 2024-10-28

**Soundness:** 3
**Presentation:** 3
**Contribution:** 3
**Rating:** 5
**Confidence:** 5

**Summary:**

This paper introduces a new supervised learning approach called Supervised Consistency Learning (SCL), which uses noise-perturbed label hints to improve predictions for complex labels. Unlike traditional methods, SCL progressively refines predictions through multi-step inference, enhancing performance in vision, text, and graph tasks by focusing on label complexity.

**Strengths:**

- This paper introduces a simple yet effective approach, showing strong performance in tasks like segmentation.
- The idea of adding noise to labels to learn consistency is a unique concept, enabling models to refine predictions progressively.
- The approach is easy to implement and can be broadly applicable across tasks involving complex labels, adding value to supervised learning methods.

**Weaknesses:**

1. The term “SCL” has already been used for supervised contrastive learning, so using a different abbreviation is recommended.

2.  While this method shows good performance on the segmentation task, it would be beneficial to explore its applicability on large-scale classification datasets with many classes, such as iNaturalist and Open Images.

3. How much longer does training take compared to standard supervised learning, in terms of both iterations and wall clock time?

4. A key limitation of this method is the increased inference time, particularly in segmentation tasks. Details on the choice of $𝑁_\tau$ during inference, time taken per inference, and wall-clock time compared to a single standard inference on a supervised model would be helpful.

5. Most results in the experimental section were shown using single-step predictions, with limited results provided for multistep predictions. Specifically, Figure 3 shows qualitative results where multistep predictions perform better than single-step, but in Table 1, only results for single-step predictions are provided, and multistep results are absent. Presenting multistep results selectively raises doubts about the performance of multistep predictions.

6. The baseline models used in the experiments are quite outdated (segmentation models from 2017-2019, N-body simulation models from 2015-2017), so it is necessary to verify whether performance improvements still occur when applied to the latest methods.

7. There is no ablation study on the proposed method in the paper. Specifically, Equation 5 uses triadic distance as a loss, but an ablation study is needed to verify whether triadic distance is an essential component. Additionally, an ablation study on the step size in the training process seems also necessary, as it needs to be demonstrated that a large step size is essential, given that only single-step or few-step predictions are actually used during inference.

8. As it takes multiple forward passes for an inference, how does performance compare to ensemble methods, that require a similar number of inferences such as Bayesian model averaging, stochastic weight averaging, or SWAG?

9. Even though the method was proposed for supervised learning with a large number of classes, I wonder what would be the performances for tasks with a relatively small number of classes, e.g. classical classification problems such as imagenet or cifar-100.

10. Definition 2.1 lacks the formal structure expected of a definition, so rephrasing it to clarify the characteristics of "complex labels" would enhance the readability.

11. In the introductory part and motivation, the authors mention information bottleneck (IB), but the derivation of the algorithm is fairly relevant to IB. More discussion or theoretical relationship between the proposed algorithm and IB would be beneficial.

If the rebuttal is satisfactory, I am willing to raise the score.

**Questions:**

See weaknesses part.

---

### Official Review · Reviewer_N41R · 2024-10-29

**Soundness:** 3
**Presentation:** 3
**Contribution:** 1
**Rating:** 5
**Confidence:** 4

**Summary:**

This paper leverages the consistency model framework to solve a variety of problems across different domains: segmentation in the image domain, the N-body problem and combinatorial optimization in the graph domain, and next-token prediction in the language modeling domain. The authors demonstrate significant performance improvements over various baselines, emphasizing the benefits of adopting the consistency model framework over traditional supervised learning approaches. The proposed Supervised Consistency Learning (SCL) framework is a novel training paradigm that introduces a progressive label reconstruction approach. Instead of directly predicting labels from input data, SCL uses a combination of single data input and a time step t to produce a noisy target, and aims to reduce the distance on the output target to ensure consistency. This approach enhances the model's ability to learn complex labels, providing a structured learning process that facilitates multi-step inference akin to gradual denoising, thereby improving prediction quality in tasks with complex labels.

**Strengths:**

1. Novelty: This paper introduces a unique and previously unexplored model. The approach of producing different outputs for the same input x based on the time step t is quite novel. While it has some similarities to Bayesian concepts, it is particularly interesting as it allows for a kind of single-model ensemble, which is an uncommon characteristic.

2. Extensive Experiments: The experiments conducted in this paper are diverse and thorough, demonstrating a high level of completeness and providing convincing support for the proposed approach.

**Weaknesses:**

1. Inconsistencies with Consistency Model Definition: The paper Claims that SCL framework is a nevel training paradigm which embodies concistency model framework, however, I don't think whether SCL framework is Consistency model. Specifically, the model does not receive noisy inputs, which contradicts the typical definition of consistency models, where the output should adapt to noisy changes in the input. Rather, the model produces **different outputs based solely on varying time steps t, similar to using t as a prior in a Bayesian model**. The framework is much close to Baysian model. The SCL framework gets image and time t, which is similar to get image and prior importance. If the prior is important, then the model should sample the output in much uncertain distribution. In this viewpoint, the sampling algorithm is just baysian inferecne and it can explain, with marginal performance improvements.

2. Inefficiency in Multistep Prediction: The proposed multistep prediction approach raises questions about its efficiency. In the described algorithm, it appears that inference is performed for all time steps t and then averaged, as illustrated in Figure 3, where time step values range from 1000 to 0. This implies nearly 1000 inferences, which seems naively similar to multiple rounds of training and ensembling. Moreover, the performance gains compared to ensemble approaches appear marginal in most cases.

3. Low Reported Performance: Upon examining Table 1, the reported results seem unusually low compared to well-known baselines reported in other prominent papers. For example, MobileNetV2 + SCL shows a significant improvement over the baseline MobileNetV2, but the reported baseline mIoU value of 17.84 seems unusually low, raising concerns about its validity, especially considering that even the 2015 FCN architecture reported an mIoU of 30. This suggests that the baseline performance reported for MobileNetV2dilated is abnormally low. Similar discrepancies are observed with other baselines compared to values reported on platforms like Papers with Code.

**Questions:**

1. In table 3 GCN section, the inference time is reduced 25s to 24s. How can this happen? as shown in Algorithm 2, it uses multistep prediction. Meaning it inferences N times more than baseline.

2. Algorithm 1 does not fit to the paper's claim, as it wants to minimise the distance between noisy target and hard target. Why does the loss want to make every $y^t$ certain? In consistency model framework, it should outputs varying confidences with time$t$

---

### Official Review · Reviewer_uQkE · 2024-11-03

**Soundness:** 2
**Presentation:** 2
**Contribution:** 2
**Rating:** 5
**Confidence:** 4

**Summary:**

The paper proposes a learning and inference paradigm aimed at both classification and regressions tasks.
The method uses a time-conditional network like classically used in diffusion and consistency models.
The network takes two inputs, the input to classify and the noisified labels, and outputs the noise free labels.
So far this is a classical setup, already used in super-resolution literature, etc...
The claimed novelty lies in the loss which (1) maps the noisified labels to the noise-free labels and (2) that two independently noisified labels for the same input should map to the same labels. This departs from time-consistency proposed in consistency models. Instead, here the consistency applies to independently noisified labels.

**Strengths:**

- The main strength of the paper is a marginal improvement on several benchmarks.
- The algorithms help understand the paper

**Weaknesses:**

- The loss does not seem well grounded
  - The $\lambda_2$-term  $\lambda_2 d(f(x,y_t,t),f(x,y_{t'},t'))$ seems redundant with the $\lambda_1$-term $\lambda_1 (d(f(x,y_t,t),y) + d(f(x,y_{t'},t'),y))$ since the $\lambda_1$-term is already mapping $f(x,.,.)$ to $y$.
  - This loss also is disconnected from the standard consistency models definition since it has not self-teaching aspect (not recurrence), which is at odds with the title "Your Consistency Model is Secretly...". What is proposed here does not look like what is classically defined as a consistency model to me. To be more explicit the $\lambda_2$-term is not recurrent since it refers to two independently sampled noise (also called independent trajectories in the paper), and unlike in consistency models there's no 'stop_gradient' indicating that one side is acting as a teacher for the other.
  - I suspect the whole loss could be simplified to $d(f(x,y_t,t),y)$, which basically is, in expectation, the case when $\lambda_2=0$.

- Lack of ablation studies
  - Authors claim that more denoising steps lead to better results but I didn't see any ablation study confirming that claim.
  - The $\lambda_1$ and $\lambda_2$ terms seem important in the loss and yet no in-depth numerical analysis of various combinations is shown, this would help me gain more insight into the necessity of the two terms for the proposed loss.

- Improvements are marginal overall: typical claimed gains are of 1 point (on various metrics and various domains) with one exception for MobileNetV2 which was originally performing a lot worse than other networks in the first place.

- The experiments cover 4 different domains leaving me with a feeling of breadth at the expense of depth.

- The writing is hard to follow in general and there's a fair amount of repetition. In particular the method could be better presented although the inclusion of the algorithm helped a lot to comprehend what is actually being proposed.

**Questions:**

1. In the continuous label-setting, how does your method compare to diffusion-based regression like typically used for super-resolution (there are plenty of them that been published: ResShift, SR3, just to name a few)?
2. Quoting from the paper "... and the model trades the output diversity to better capture y. This calls for the requirement of consistency
extending across all trajectories ...". Why? What happens when you stick to the classical time-consistency definition from consistency models? I would be interested in seeing this claim actually substantiated by numerical results.
3. This is somewhat redundant with my points on weaknesses, namely the lack of ablations, but what happens when setting $\lambda_2=0$? I would really love to see an ablation on this.

---

### Official Review · Reviewer_1Fq1 · 2024-11-08

**Soundness:** 3
**Presentation:** 3
**Contribution:** 2
**Rating:** 8
**Confidence:** 4

**Summary:**

This paper explores an alternative method of doing supervised learning. The motivation is that, even though supervised learning has been largely successful in machine learning, when the label distribution is very complex (e.g. graphs and pixels),  learning the posterior distribution can be challenging. In this work, the author proposed learning a consistency model for the label distribution given the data distribution and noisy label samples. Through extensive experiments, including semantic segmentation, n-body simulation, combinatorial optimization and next-token prediction, the paper shows that their method consistently performs better without a large sacrifice in inference time compared to their discriminative modeling counterparts.

**Strengths:**

This work has a fair amount of novelty, in the sense that consistency is still a relatively new generative model, and there isn’t much work on modeling the target distribution. As for the writing, the proposed method is fairly straight-forward, as it is applying consistency model to the label distribution conditioned on the data distribution, rather than the typical setting of applying it to the data distribution itself. The introduction of consistency model, multi-step inference, and forward steps in categorical and continuous distributions are clearly defined, making the paper self-contained and providing sufficient information for the reader to follow through.  For experiments, the authors have done a fair job in testing their method in multiple settings, such as semantic segmentation, n-body simulation, combinatorial optimization and next-token prediction. Each setting is also provided a fair number of baselines and comparisons to clearly demonstrate that their method can perform better than current state-of-the-art. Overall, the method is sound and works reasonable well within expectations.

**Weaknesses:**

1.	One shortcoming is that the paper assumed the data provides sufficient information about the labels, but this often happens when there is an abundance of data. It is an assumption that is implicitly made by the method, but was not mentioned at all.
2.	The writing of the paper begins and motivates from the perspective of Information Bottleneck. However, from the writing, it’s not exactly clear how the high level intuition connects with the assumptions about the model. Intuitively, I believe the authors are trying to argue that consistency models also make the assumption that the distribution it’s trying to model (i.e. the target distribution here) is also trying to learn a low-dimensional representation, for instance. However, this is not explicitly said or mentioned. In other words, the discussion on IB seems to distract from the discussion rather than helping the discussion. The introduction would have read fine without any mentioning of IB.  Some additional comments regarding the connection in the related work section would make the motivation of the work more clear and prominent.
3.	While one could argue that consistency models are a recent rising trend, the method is somewhat simply applying consistency models in place of other generative models for learning target distributions. In this sense, there is limited novelty. It can help if the authors can comment more on how consistency model achieves something that previous generative models cannot. This weakness is also listed as a question for the authors below also.

**Questions:**

1. Can the authors provide a justification for why this cannot be done with other generative models? In other words, what does consistency model offer as an advantage that other generative models (such as diffusion models or VAEs) cannot do?
2. Can the authors provide some comments on how the method would perform in the case where there is class imbalance, or an extremely large number of classes (e.g. extreme classification settings)? Similarly, can the authors provide some comments also in the case there is domain shifts, or out-of-domain issues?

---

### Public Comment · ~Junlong_Huang2 · 2024-11-25
**The Authors Do Miss an Important Reference**

I believe there is a paper closely related to this submission, but the authors have not cited it in their current submission version so far:
> Li, Alexander C., et al. "Your diffusion model is secretly a zero-shot classifier." Proceedings of the IEEE/CVF International Conference on Computer Vision. 2023.

I think the authors must have read this paper, which can be inferred from their own title, as it is quite similar, so they should not have missed it.

---

> ### Author Response · Authors · 2024-11-25
>
> Thanks for your focus on our paper. However, we kindly ask that you carefully review our methodology before posting such comments. Paper [1] is fundamentally different from ours. **[1] focuses on the application of pre-trained text-to-image diffusion models, while we propose a novel training paradigm for prediction tasks.** We are unclear on how Paper [1] relates to our supervised consistency learning framework, aside from similarities in title structure. Indeed, titles of this structure are common in the literature (e.g., [2, 3, 4, 5, 6, 7, 8]), and we did not reference [1] at all during the design of our methodology.
>
>
> For your information, the summary of these two works are as follows. **What [1] does:**
> [1] utilizes a pre-trained text-to-image diffusion model for classification by measuring the matching degree between the input image and different label descriptions, selecting the most matched one as the label. **What this paper does:** Our method leverages the concept of consistency modeling to enhance the traditional supervised learning framework.  Specifically, we learn the mapping of `data input x, noised label y_t (as additional hint) -> full label y` instead of `data input x -> full label y`, and then enforce this mapping to be consistent across different noise levels within the label space, i.e., $y_t, 0\leq t\leq T$.
>
>
> If you believe there is a relevant connection, we would appreciate it if you could clarify the relationship between the two works, perhaps by referring to specific topics in the Related Work section. This would help us assess whether a citation is warranted.
>
> [1] Your diffusion model is secretly a zero-shot classifier. CVPR 2023.
>
> [2] Your GAN is secretly an energy-based model and you should use discriminator driven latent sampling. NeurIPS 2020.
>
> [3] From r to Q∗: Your Language Model is Secretly a Q-Function. COLM 2024.
>
> [4] Your classifier is secretly an energy based model and you should treat it like one. ICLR 2020.
>
> [5] Direct preference optimization: Your language model is secretly a reward model. NeurIPS 2024.
>
> [6] Your ViT is Secretly a Hybrid Discriminative-Generative Diffusion Model. Arxiv 2022.
>
> [7] Your Weak LLM is Secretly a Strong Teacher for Alignment. Arxiv 2024.
>
> [8] Your Large Language Model is Secretly a Fairness Proponent and You Should Prompt it Like One. Arxiv 2024.

---

> > ### Public Comment · ~Junlong_Huang2 · 2024-12-02
> >
> > Thank you for the authors' responses. I would like to further clarify that, as the authors have stated, the paper I supplemented can also perform classification tasks. The original paper specifically discusses supervised learning and compares their method with supervised learning methods. Moreover, the consistency model used in this paper is closely related to diffusion. However, the writing intent in the previously submitted version seemed to suggest that the authors of this submission were the first to use a diffusion-based model for supervised learning. This is why I say this: because the authors have not mentioned at all that other previous diffusion-based methods have already been able to solve some of the tasks conducted in the experimental part of this submission.
> >
> > Therefore, I believe the authors should particularly not omit the paper I mentioned. Moreover, the title of the paper in the previous version was very similar to the one I supplemented (although I have noticed that the authors have now renamed it in the new PDF file). And since the authors have listed a considerable number of paper titles in the format of "your ... is secretly...", it further indicates that the authors are likely to be aware of the paper I supplemented. Thus, I think the authors should not omit it, should not fail to mention this work at all, and should not fail to give credit to those who have used diffusion-based methods to perform tasks similar to this submission before, such as the basic classification. Besides, as for the semantic segmentation task that the authors have conducted, the authors of the paper I supplemented mentioned in their paper that the following two also use diffusion models for semantic segmentation tasks, but the authors have not cited these two papers either:
> >
> > > 1. Baranchuk, Dmitry, et al. "Label-Efficient Semantic Segmentation with Diffusion Models." *International Conference on Learning Representations*.
> >
> > > 2. Clark, Kevin, and Priyank Jaini. "Text-to-image diffusion models are zero-shot classifiers." *Proceedings of the 37th International Conference on Neural Information Processing Systems*. 2023.
> >
> > Additionally, I have noticed that two reviewers have raised their scores, one from 6 to 8, and one from 3 to 5. Therefore, it can be inferred that the authors should have responded to the reviewers' comments, but I cannot see the responses. **Since the ICLR rebuttal is default public to everyone, I believe it should be that the authors deliberately did not set the responses to be public to everyone.** As ICLR is a conference with a very open and transparent review stage, most authors would choose to discuss openly with reviewers, but I am not clear about the intentions of the authors in doing so.

---

> > > ### Author Response · Authors · 2024-12-02
> > >
> > > Thanks for your continuous focus. We want to emphasize that we never intended to deliberately omit related works, and we firmly agree with you that giving credit to previous foundations of the current work is very important. **As mentioned in our previous comment, the reason we have not included paper [1] in our related work section is that we believe it belongs to a fundamentally different methodological category for different tasks, with no significant methodological similarity to our method, aside from both being somehow related to diffusion models.** However, our only connection to diffusion models lies in the noising process within the label space, and we did not reference the core design of diffusion models, i.e., their denoising learning procedures at all. Upon closer examination of the algorithmic details, it can be discovered that our method is more of an enhancement to traditional supervised learning by incorporating the consistency mapping concept. Additionally, our approach does not focus on or conduct any experiment on basic classification tasks as in paper [1], but is designed for supervised learning tasks involving complex labels. The motivation behind our method comes from the information bottleneck theory of supervised learning, and our prediction consistency learning process is specifically designed to better capture the intricate information encoded in complex labels.
> > >
> > > Please kindly recall that **what this paper does** is leverage the concept of consistency modeling to enhance the traditional supervised learning framework from `data input x, noised label y_t (as additional hint) -> full label y` instead of `data input x -> full label y`, and then enforce mapping consistency. **What paper [1] proposes** is an application of pre-trained text-to-image diffusion models for zero-shot classification by measuring the matching degree between the input image and different label descriptions with the pre-trained diffusion model. This approach does not involve any supervised training, which is why the title of paper [1] refers to it as a zero-shot classifier.
> > >
> > > We would like to clarify more specifically that this paper does not fall into the category of consistency model applications (let alone diffusion models, since the consistency mapping concept we refer to is unrelated to prior diffusion models) to prediction tasks, and the primary baseline of this paper is the traditional supervised training procedure. We acknowledge that the original title, "Your consistency model is secretly...," may have led to confusion about whether our paper applies consistency models directly to supervised learning tasks. As the reviewers pointed out, the method we propose is no longer a generative consistency model within the supervised learning framework. In response to this feedback, we have revised the title to "Prediction consistency enhances supervised learning..." as suggested by the reviewers.
> > >
> > > Regarding the visibility of the responses, please note that ICLR allows authors to restrict the visibility of replies, as previously notified in the email. We have restricted the comments to reviewers and chairs based on the original ratings of this paper, as we hope to reserve new discussions and results for potential future resubmissions. We are open to making our responses publicly available if requested by the chairs or reviewers.
> > >
> > >
> > > [1] Your diffusion model is secretly a zero-shot classifier. CVPR 2023.

---

### Meta-Review · Area_Chair_oTon · 2024-12-20

**Metareview:**

This paper proposes a supervised training paradigm using "diffusion-like" process. Instead of directly mapping inputs to labels, the authors introduce “noised” label hints and enforce prediction consistency across different noise levels.


The authors’ rebuttal helped clarify some points and added ablations, but it did not sufficiently alleviate the skepticism around conceptual alignment, practical utility, and the breadth of experimentation. This paper can be further benefit from rigorous theoretical treatment, stronger baselines, and more robust comparative evaluations before acceptance.

**Additional Comments On Reviewer Discussion:**

During the rebuttal period, the reviewers and authors engaged in a detailed discussion. Here is a summary of the key points raised and how the authors addressed them:

1. **Connection to Consistency Models and Terminology (Raised by Reviewer uQkE, Reviewer N41R, Reviewer zvSo):**
   - Multiple reviewers expressed confusion about whether the proposed method truly aligns with the classical definition of consistency models used in generative tasks. For example, *Reviewer uQkE* and *Reviewer N41R* pointed out that the method does not exactly implement the same self-teaching recurrence or noising processes as standard consistency models.
   - The authors responded by clarifying that their approach is inspired by the concept of consistency modeling, but adapted to supervised learning
   - While this clarification was appreciated, *Reviewer N41R* and Reviewer zvSo remained skeptical about how strongly the proposed method can be considered a consistency model under the traditional definition.

2. **Ablation Studies and Incremental Improvements (Raised by Reviewer uQkE, Reviewer  zvSo, Reviewer N41R):**
   - Several reviewers, including *Reviewer uQkE* and *Reviewer  zvSo*, requested more thorough ablation studies to understand the necessity and effectiveness of each component of the loss, as well as the effect of multi-step inference. The authors provided additional ablation results for the n-body simulation task, and included more comparisons of multiple denoising steps.
   - While these extra experiments offered some clarity, Reviewer uQkE noted that the improvements from certain terms were minor and could be explained by regularization or augmentation effects. Similarly, *Reviewer zvSo* felt that while the improvements were present, they were often marginal, and that deeper or more rigorous analysis could strengthen the paper’s claims.

3. **Choice of Baselines and Benchmark Quality (Raised by Reviewer  N41R, Reviewer zvSo):**
   - Reviewers questioned the reliance on relatively outdated baselines and certain experimental setups that produced low performance numbers compared to commonly reported state-of-the-art results (e.g., segmentation performance on ADE20K). The authors explained that they aimed for a fair internal comparison across multiple tasks, consistently applying their method.
   - However, *Reviewer N41R* remained unconvinced, emphasizing that demonstrating gains over stronger and more modern baselines would bolster the case for the method’s general utility. *Reviewer ZvSo* likewise suggested testing on more large-scale classification tasks or well-established benchmarks for broader credibility.

4. **Relation to Information Bottleneck and Theoretical Foundations (Raised by Reviewer zvSo):**
   - The introduction and motivation mentioned information bottleneck (IB), but *Reviewer zvSo* found the derivation and theoretical connection to IB insufficient. The authors provided a conceptual explanation, noting how their method aims to handle complex labels by progressively introducing label information, akin to addressing predictive bottlenecks that IB identifies.
   -  However, the reviewer felt the response did not fully alleviate the lack of a formal theoretical treatment linking IB principles to the proposed training procedure.



By the end of the discussion, the major concerns still remain including the limited theoretical rigor connecting to IB, the modest and sometimes niche improvements, and the absence of strong benchmarks or modern baselines.

---

### Decision · Program_Chairs · 2025-01-22

Reject